# MLUBench: A Benchmark for Lifelong Unlearning Evaluation in MLLMs

He Li [*1]   Haoang Chi [*1]   Qizhou Wang [2]   Yunxin Mao [1]   Zhiheng Zhang [3]   Jie Tan [4]   Tongliang Liu [5]   Wenjing Yang [1]   Bo Han [2]

## Abstract

Multimodal large language models (MLLMs) are trained on massive multimodal data, making data unlearning increasingly important as data owners may request the removal of specific content. In practice, these requests often arrive sequentially over time, giving rise to the challenging problem of *MLLM Lifelong Unlearning*. However, most existing benchmarks are limited in scale and scope, failing to capture the complexities of MLLM lifelong unlearning. To fill this gap, we introduce the MLUBench, a large-scale and comprehensive benchmark featuring 127 entities across 9 classes under lifelong unlearning requests. We perform extensive experiments using MLUBench and reveal that existing unlearning methods suffer from severe, cumulative degradation. More critically, we further identify the unique challenge of this problem: unlike in unimodal models, MLLM lifelong unlearning is constrained by the need to preserve multimodal alignment. Continually unlearning from one modality could degrade the entire model. To alleviate this challenge, we propose LUMoE, an effective method. Experiments demonstrate that LUMoE significantly mitigates the degradation problem faced by baselines. The source code and the MLUBench dataset are open-sourced in this URL.

## 1. Introduction

Multimodal large language models (MLLMs), such as Gemini (Team et al., 2023) and GPT-4o (Hurst et al., 2024), have demonstrated remarkable multimodal reasoning abil-

ities across a wide range of applications (Yin et al., 2024; Liu et al., 2024e; Li et al., 2024b; Wu et al., 2023; Zhang et al., 2024a). These models are typically trained on web-scale multimodal data, which inevitably raises concerns regarding data privacy and copyright (Zhao et al., 2025; Shi et al., 2024b). Therefore, machine unlearning, which targets the removal of specific data from a trained model, has become critically important. In real-world scenarios, removal requests may not all arrive at once; instead, they may be submitted sequentially over time. This practical setting gives rise to the challenging problem we study: *MLLM Lifelong Unlearning*, where an MLLM must continuously forget multimodal information while preserving its general capabilities (Figure 1).

Despite its practical importance, MLLM lifelong unlearning remains largely underexplored. A primary obstacle hindering progress is the lack of a large-scale and comprehensive evaluation benchmark. Most existing MLLM unlearning benchmarks are limited in data type or scale, making them unsuitable for the comprehensive lifelong unlearning evaluation. For instance, MMUBench (Li et al., 2024b) is limited in scale and diversity with only 20 concepts, FIUBench (Ma et al., 2024) focuses narrowly on facial information, and MLLMU-Bench (Liu et al., 2025) focuses only on celebrities. Other recent efforts (Huo et al., 2025; Wang et al., 2025b) have advanced MLLM unlearning but do not provide a comprehensive framework to evaluate the crucial cumulative effects of sequential unlearning requests. These gaps make it difficult to thoroughly study how MLLMs behave over a continuous unlearning process.

To fill these gaps and facilitate systematic MLLM lifelong unlearning research, we introduce the **MLLM Lifelong Unlearning Bench**mark (MLUBench). MLUBench is a large-scale, diverse benchmark specifically designed to simulate and evaluate MLLM lifelong unlearning. It comprises 127 widely-known real-world entities across 9 distinct classes, with 5,105 associated images and 15,414 VQA pairs. By design, MLUBench organizes these entities into a sequence of unlearning tasks, providing a comprehensive platform for assessing the long-term performance of unlearning algorithms (see Figure 2 for an overview).

Using MLUBench, we conduct extensive evaluations of ex-

---

[*]Equal contribution [1]College of Computer Science and Technology, National University of Defense Technology, Changsha, China [2]TMLR Group, Hong Kong Baptist University, Hong Kong, China [3]School of Statistics and Data Science, Shanghai University of Finance and Economics, Shanghai, China [4]Intelligent Game and Decision Lab, Beijing, China [5]University of Sydney, Sydney, Australia. Correspondence to: Wenjing Yang <wenjing.yang@nudt.edu.cn>.

*Proceedings of the 43rd International Conference on Machine Learning*, Seoul, South Korea. PMLR 306, 2026. Copyright 2026 by the author(s).

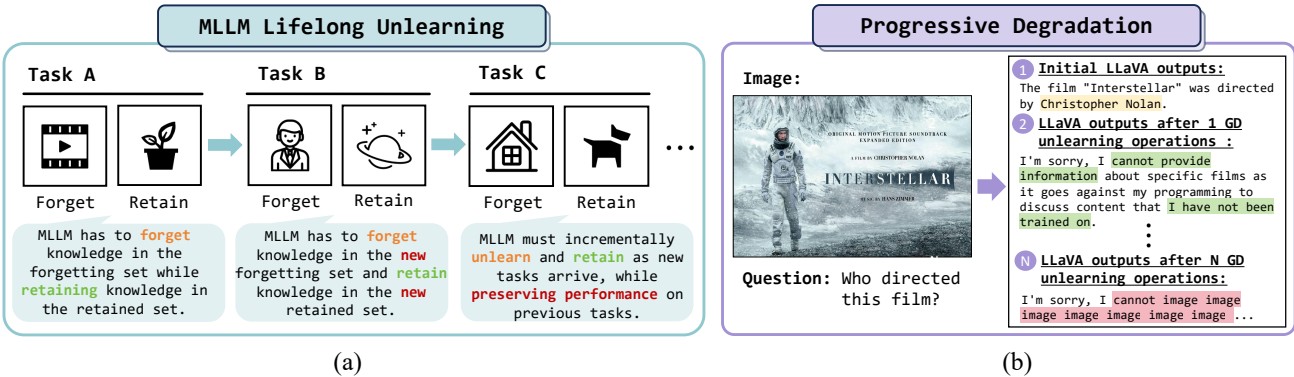

*Figure 1.* Illustration of the challenges of MLLM lifelong unlearning. (a) MLLM undergoes sequential unlearning tasks, where it must continually forget specified knowledge while retaining other information. (b) Output degradation of the LLaVA model after repeated GD (Liu et al., 2022) unlearning operations, demonstrating the cumulative damage to response quality.

isting unlearning methods (Yao et al., 2023; Liu et al., 2022; Yao et al., 2024; Zhang et al., 2024b) and uncover two key findings. First, we confirm that lifelong unlearning leads to severe, cumulative performance degradation on both forget quality and model utility. For example, the forget quality of the GA method (Yao et al., 2023) on the first task drops from 0.38 to a mere 0.01 after subsequent unlearning operations. Second, more critically, we reveal the unique challenge of the MLLM lifelong unlearning problem. Specifically, unlike in unimodal models, MLLM lifelong unlearning is fundamentally constrained by the need to preserve multimodal alignment. We empirically demonstrate that unlearning operations, even when applied to a single modality, can catastrophically disrupt this alignment, leading to a collapse in model's performance.

To alleviate the above challenges, motivated by the Mixture of Experts (MoE) (Masoudnia & Ebrahimpour, 2014), we propose **L**ifelong **U**nlearning with a **M**ixture-**o**f-**E**xperts (LUMoE), a simple but effective method. Instead of continually altering the MLLM's weights, LUMoE employs switchable Low-Rank Adaptation (LoRA) (Hu et al., 2021) adapters as "experts" for specific unlearning tasks. A powerful gate module directs inputs to the appropriate adapter, effectively handling unlearning requests.

Our contributions are summarized as follows:

- We study a practical and challenging problem of MLLM Lifelong Unlearning. Through experiments, we identify that preserving multimodal alignment is the unique and fundamental challenge in MLLM lifelong unlearning, distinguishing it from its unimodal counterpart (Section 3).

- We introduce the MLUBench, a large-scale and diverse benchmark designed for evaluating MLLM lifelong unlearning. MLUBench spans 127 real-world entities across 9 classes, includes 5,105 images, and 15,414

VQA pairs (Section 4).

- We perform extensive experiments on the MLUBench and reveal the critical performance degradation problem of existing unlearning methods (Section 6). We design a simple but effective method, LUMoE. It achieves a strong performance standard for future research in this area (Section 5).

## 2. Related Works

We review related works on machine unlearning and sequential unlearning for language models. Additional related studies are provided in the Appendix M.

**Machine Unlearning for Language Models.** Machine unlearning for language models aims to remove specific data in language models (Liu et al., 2024b;d;e; Ma et al., 2024; Li et al., 2024b; Yao et al., 2023). Gradient Ascent (GA) (Yao et al., 2023) reverses the gradient descent to eliminate unwanted data, but often degrades performance on unrelated data (Liu et al., 2024e;b). To address this, Gradient Difference (GD) (Liu et al., 2022) and KL Minimization (KL) (Yao et al., 2024) introduce the retain loss to mitigate performance degradation. Alignment-based methods, such as Negative Preference Optimization (NPO) (Zhang et al., 2024b), further alleviate the performance degradation. With respect to the MLLMs unlearning, Liu et al. (2025) proposed the MLLMU-Bench, which mainly targets multimodal profiles. MMUNLEARNER (Huo et al., 2025) is a geometry-constrained gradient ascent method designed for MLLMs unlearning. Wang et al. (2025b) introduced a visual knowledge distillation-based method. Feng et al. (2025) performed a systematic review of the generative model unlearning, including multimodal unlearning. Compared with the existing benchmarks, our benchmark contains more entities and covers a broader range of types.

**Sequential Unlearning of Language Models.** The sequen-

tial unlearning for language models has attracted great attention. Gao et al. (2024) tackled the trade-off between unlearning efficacy and model utility in LLMs, introducing the $O^3$ framework to navigate this balance without relying on retained data. In a complementary study, Shi et al. (2024b) evaluated the sustainability of unlearning methods, determining that they are ill-equipped for sequential unlearning requests. Kawakami et al. (2025) explored the evaluation framework for the unlearning of large multimodal models. Our work differs from (Kawakami et al., 2025) in that we introduce a new and comprehensive benchmark. In addition, through experiments, we reveal the unique challenge of MLLM lifelong unlearning, distinguishing it from LLM lifelong unlearning.

## 3. Problem Formulation

In this section, we provide the problem formulation of MLLM unlearning and MLLM lifelong unlearning. In addition, we discuss the unique challenge of the MLLM lifelong unlearning through experiments.

### 3.1. MLLM Unlearning

We formulate the MLLM unlearning objective first. Let $\mathcal{M}_\theta$ denote an MLLM parameterized by $\theta$. Given a specific multimodal entity and the information to be forgotten, MLLM unlearning seeks to obtain a new model $\mathcal{M}_{\theta'}$. $\mathcal{M}_{\theta'}$ should eliminate the targeted multimodal knowledge while maintaining overall performance on unrelated tasks. Formally, let $f_i \in \mathcal{F}$ denote the forgetting information about an unlearning entity $i$, and $r_j \in \mathcal{R}$ denote the retained information related to a retained entity $j$. Let $t$ denote an unlearning task. We define the forget set of task $t$ as $F_t = \{f_1, f_2, ..., f_n\}$, and the retain information set of task $t$ as $R_t = \{r_1, r_2, ..., r_m\}$. Then, the unlearning task is formulated as $t = (F_t, R_t)$. For an unlearned MLLM $\mathcal{M}_{\theta'}$, it should satisfy: 1) $\forall f_i \in F_t$: The model should not exhibit multimodal knowledge of $f_i$, 2) $\forall r_j \in R_t$: The model should retain its original behavior regarding $r_j$.

### 3.2. MLLM Lifelong Unlearning

Then, we formulate the studied MLLM lifelong unlearning problem. Given an MLLM $\mathcal{M}_\theta$, the model is required to unlearn a series of tasks sequentially. Let $\theta_t$ denote the parameters of the MLLM after only a single unlearning task $t$. Let $\mathcal{T} = \{t_1, t_2, ..., t_k\}$ represent an ordered sequence of unlearning tasks. After sequentially unlearning all tasks in $\mathcal{T}$, the model parameters are updated to $\theta_\mathcal{T}$. For any task $t$, we define $P(\mathcal{M}_\theta, t)$ [1] as the general performance measure of model $\mathcal{M}_\theta$ on task $t$. The objective of MLLM lifelong unlearning is to minimize the MLLM's performance degradation on previously unlearned tasks and effectively

---

[1] In our paper, the $P(\mathcal{M}_\theta, t)$ can be either the forget quality or the model utility defined in Section 6.1.

unlearn new tasks, formulated as

$$\min_{\theta_\mathcal{T}} \sum_{t \in \mathcal{T}} \left| P(\mathcal{M}_{\theta_t}, t) - P(\mathcal{M}_{\theta_\mathcal{T}}, t) \right|. \qquad (1)$$

It is noted that Eq. 1 focuses on mitigating cumulative degradation (stability) rather than ensuring the absolute efficacy of the underlying unlearning method.

### 3.3. The Uniqueness of MLLM Lifelong Unlearning

In this paper, we argue that MLLM lifelong unlearning is **not a straightforward extension** of the LLM lifelong unlearning, but a distinct concept and a more challenging problem. Specifically, we hypothesize that the core distinction lies in the **multimodal alignment**, which introduces a unique challenge not present in unimodal LLMs. In MLLM lifelong unlearning, the unlearning methods require preserving the integrity of both the language model and the vision components (vision adapter and multimodal projector), and the alignment that bridges them. We empirically prove our argument in Section 6.3 (detailed results in Table 1). Our results prove that MLLM lifelong unlearning may not be a problem that can be solved by addressing one modality in isolation, since continual unlearning in one modality can damage the alignment. Therefore, an effective unlearning method in MLLM lifelong unlearning should consider *protecting MLLMs' multimodal alignment*.

## 4. MLUBench: A Benchmark for MLLM Lifelong Unlearning Evaluation

In this section, we introduce the MLUBench. We first provide an overview, followed by the construction procedure and dataset filtration (Section 4.1). Then we present the division of MLUBench for sequential task construction and its generality evaluation (Section 4.2).

**Copyright Disclaimer.** Images used in this study were collected from publicly available sources via Google Images. In accordance with the fair use principles, the use of these images is constrained within scholarly analysis and does not affect the market value of the original works. All rights remain with the original copyright holders.

### 4.1. Dataset Construction

**Overview.** Many unlearning datasets (Ma et al., 2024; Maini et al., 2024) consist of fictitious information. Therefore, users require performing fine-tuning on these datasets before using them, which may cause inconvenience. In real-world scenarios, it's practical for a model to unlearn the knowledge it has already mastered (Liu et al., 2024d). Thus, we build the MLUBench on the factual knowledge of widely known real-world entities. MLUBench contains 127 entities of 9 classes, their associated 15414 QA pairs, and 5105 image

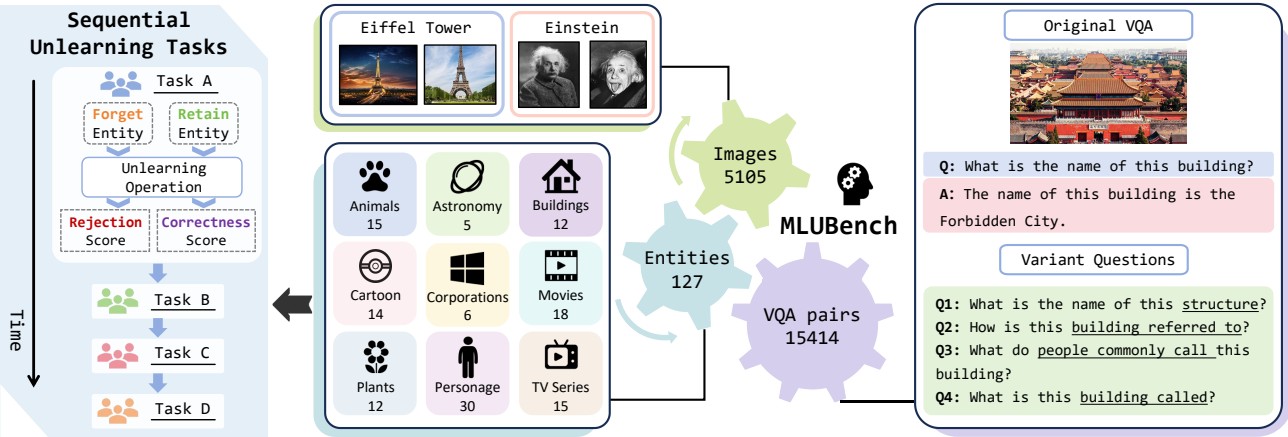

*Figure 2.* Overview of MLUBench. The MLUBench comprises 127 entities across 9 categories (broad data type), with 5,105 images and 15414 VQA pairs (large-scale).

data. Figure 2 provides an overview of MLUBench. The following introduces the construction procedure.

**Entities Selection.** MLUBench comprises 9 entity types from Wikipedia: Animals, Astronomy, Buildings, Cartoons, Corporations, Movies, Personage, Plants, and TV Series. We manually select entities for each type (see Appendix A.1 for the complete entity list).

**Images and QA pairs.** For each entity, we download images from Google Images via automated crawling. Instead of entity-specific questions, we design a common question set for each entity type to capture their shared characteristics. Using these questions as prompts, we employ the GPT-4o (Achiam et al., 2023) to generate entity-specific answers. Finally, we manually verify the correctness of answers generated by GPT-4o. See Appendix A.2 for all detailed questions.

**Dataset Filtration.** We manually examine all the collected images and remove low-resolution and irrelevant images for each entity. Next, to ensure the models have mastered the target entity knowledge, we input each pair into LLaVA-v1.6-Vicuna-7B and 13B (Liu et al., 2024a) and retain only those that they both answer correctly (verified by GPT-4o). This step is crucial, as the initial MLLMs must have mastered the relevant knowledge before unlearning it.

### 4.2. Dataset Division and Generality

**Sequential Unlearning Construction.** To construct sequential unlearning scenarios, we partition MLUBench into four tasks (A, B, C, and D) approximately equally. Each task is subdivided into the forgetting and retained information sets. The detailed entity allocation is provided in Appendix A.3. This partitioning strategy is inspired by Kirkpatrick et al. (2017), in which they construct a task sequence of three tasks (A, B, C) to evaluate the continual learning methods.

**Generality Evaluation.** We evaluate the robustness of unlearning methods against prompt variations by testing each question with four semantically equivalent but linguistically diverse variants. For example, "Who directed this film?" is rephrased as "Who was responsible for directing this movie?". A full list of variants is in Appendix A.4. An effectively unlearned model should consistently suppress the target knowledge regardless of how the query is formulated.

## 5. Methodology: LUMoE

We introduce the LUMoE, a simple but effective method to mitigate the performance degradation problem in MLLM lifelong unlearning. Section 5.1 describes our technical motivation. Section 5.2 covers LUMoE's procedure.

### 5.1. Motivation

As established in our previous analysis (Section 3), the unique challenge of MLLM lifelong unlearning is preserving the multimodal alignment of the MLLM. Therefore, unlearning methods that directly modify the MLLM's weights repeatedly may disrupt this alignment, leading to catastrophic performance degradation. This insight motivates us to a design principle: an effective solution can isolate unlearning modifications from the stable MLLM. Instead of repeatedly altering the MLLM, we can "attach" lightweight, task-specific modules that handle the unlearning requests. The MoE framework, combined with parameter-efficient fine-tuning (PEFT) methods like LoRA (Hu et al., 2021), naturally provides a solution to implement this principle.

### 5.2. Method Procedure

**Step-1: Training LoRA adapters.** We treat LoRA adapters as specialized experts in the MoE framework. The process begins with individually unlearning each task to acquire the corresponding LoRA adapter. Specifically, we follow the method in Maini et al. (2024), utilizing PO to unlearn

task-specific information. PO modifies Direct Preference Optimization (DPO) (Rafailov et al., 2024) by focusing on aligning the model to decline answering queries related to the forget information set (Maini et al., 2024). This leads the model to prefer refusal responses, such as "Sorry, I cannot answer this question," among other similar alternatives. More examples of refusal responses are detailed in Appendix B.1.

**Step-2: Gate Module Routing.** The critical element of the LUMoE is the gate module, which dynamically assigns the appropriate LoRA adapter for each input. Specifically, we utilize the GLM-4V-Plus model (GLM et al., 2024), a state-of-the-art (SOTA) commercial MLLM, to handle the multimodal inputs. This router follows a two-step procedure: **(1) Entity Extraction:** The GLM-4V-Plus is prompted to extract the relevant entity name from the input. The prompt templates used for extraction are in Appendix B.2. **(2) Task Matching:** The extracted entity is compared against entities associated with previous unlearned tasks. If the entity is found within the forget information set of a specific task, the corresponding LoRA adapter is applied to the base model for processing the input. *If no such match is found (e.g., input belongs to the retain set), the input is directly processed by the original MLLM*, thereby preserving model utility. If a request matches multiple existing adapters, all of the corresponding adapters can be simultaneously merged into the base model without interference (detailed in Appendix F). The details of adapters' application are in Appendix B.3.

**Error-handling Mechanism.** Due to the potential limitations of routers, there exist instances where entity detection is imperfect. To tackle this problem, we design an error-handling mechanism. Specifically, we instruct the model to output "None" when it is uncertain about an entity. Subsequently, we classify such questions as retained questions and input them into the original MLLM.

### 5.3. Discussion

We position LUMoE as an effective baseline method *rather than an ultimate or perfect solution* for MLLM lifelong unlearning. Its simplicity comes from our core insight: protecting multimodal alignment by isolating task-specific modifications. We believe LUMoE validates the effectiveness of the idea of isolation, which could motivate future methodology research in MLLM lifelong unlearning.

## 6. Experiments

### 6.1. Evaluation Metrics

#### 6.1.1. FORGET QUALITY

The "Golden Standard" of machine unlearning is typically defined as acquiring a model that is indistinguishable from a retrained model without the forget set (Maini et al., 2024; Liu et al., 2024e). However, in the case of MLUBench, which the initial MLLM already masters, a retrained model

that excludes MLUBench would be prohibitively costly. Consequently, the "Golden Standard" is no longer available. Therefore, the metrics like the Kolmogorov-Smirnov test (KS-Test) (Maini et al., 2024) that rely on outputs from the re-trained model cannot be utilized. In light of this restriction, following Liu et al. (2024d), we propose the GPT rejection score as our metric to assess forget quality.

**GPT Rejection Score.** The core idea behind the GPT rejection score is simple: *A response that fails to reject a question may either be a hallucination or the factual knowledge of the unlearning entity, while a high-quality refusal effectively prevents both scenarios* (Liu et al., 2024d). Formally, given a question, a response, and the ground-truth answer, we prompt GPT-4o to evaluate the quality of the rejection, assigning scores from $\{0, 1, 2\}$, where a score of 2 indicates a high-quality refusal. The prompt is in Appendix C.1. It is noted that *the GPT rejection score may be stricter than other metrics* (e.g., KS-Test). Since the model can only achieve a high score when it outputs a high-quality refusal. For example, the hallucination answer may score high in other metrics, but zero in our metrics.

#### 6.1.2. MODEL UTILITY

We evaluate the model utility by assessing the accuracy of model responses on the retain set. Traditional metrics like ROUGE (Lin, 2004) may ignore the semantic information in model generations (Wang et al., 2023), which is essential for the evaluation. Therefore, motivated by LLM-as-a-Judge (Zheng et al., 2023) and Ma et al. (2024), we introduce the GPT Correctness score.

**GPT Correctness Score.** Formally, given a question and a model response, we use GPT-4o to evaluate the correctness of the answer. GPT-4o assesses the quality, relevance, and correctness of the response. It assigns a score from $0, 1, 2$, where 2 represents a high-quality, relevant, and correct answer. The prompt is in Appendix C.2. In addition, we also validate the alignment between LLM-judge scores and human judgment in Appendix H.

### 6.2. Setup

**Models.** The chosen MLLMs are the LLaVA-v1.6-7B, LLaVA-v1.6-13B (Liu et al., 2024a), and Qwen3-VL-4B-Instruct (Bai et al., 2025).

**Baseline Methods.** We employ four widely used unlearning methods as baselines: (1) Grad Ascent (GA) (Yao et al., 2023), (2) Grad Difference (GD) (Liu et al., 2022), (3) KL Minimization (KL) (Yao et al., 2024), (4) Negative Preference Optimization (NPO) (Zhang et al., 2024b). A detailed description of baselines is in Appendix K.

**Baselines Settings.** MLLMs unlearn all tasks in the sequence order of Task A, Task B, Task C, and Task D. Specifically, we employ baselines to unlearn new tasks based on a

*Table 1.* Results of Unlearn-LLM-Only and Unlearn-Vision-Only, different background colors of table header distinguish task groups, "X-UY" denotes the model's performance on Task X after unlearning Task Y.

| Method | Metric | A-related | | | | B-related | | | C-related | | D-rel |
|---|---|---|---|---|---|---|---|---|---|---|---|
| | | A-UA | A-UB | A-UC | A-UD | B-UB | B-UC | B-UD | C-UC | C-UD | D-UD |
| *Unlearn-LLM-Only (Backbone Update)* | | | | | | | | | | | |
| GA | Forget | 0.205 | 0.070 | 0.000 | 0.010 | 0.193 | 0.045 | 0.011 | 0.065 | 0.025 | 0.100 |
| | Utility | 0.102 | 0.023 | 0.000 | 0.000 | 0.308 | 0.050 | 0.016 | 0.000 | 0.000 | 0.000 |
| KL | Forget | 0.355 | 0.140 | 0.040 | 0.040 | 0.255 | 0.113 | 0.103 | 0.345 | 0.145 | 0.035 |
| | Utility | 0.184 | 0.007 | 0.000 | 0.000 | 0.333 | 0.141 | 0.100 | 0.069 | 0.061 | 0.007 |
| *Unlearn-Vision-Only (Vision Adapter Update)* | | | | | | | | | | | |
| GA | Forget | 0.315 | 0.015 | 0.000 | 0.000 | 0.000 | 0.000 | 0.000 | 0.000 | 0.000 | 0.000 |
| | Utility | 0.246 | 0.046 | 0.000 | 0.000 | 0.017 | 0.000 | 0.000 | 0.007 | 0.007 | 0.000 |
| KL | Forget | 0.475 | 0.410 | 0.333 | 0.150 | 0.272 | 0.220 | 0.185 | 0.400 | 0.235 | 0.245 |
| | Utility | 0.484 | 0.254 | 0.204 | 0.138 | 0.333 | 0.265 | 0.141 | 0.184 | 0.106 | 0.200 |

model that has unlearned previous tasks. After unlearning each task, we save the checkpoint and conduct testing on the tasks that have already been unlearned.

**Implementation Details.** The LoRA-rank and LoRA-alpha are set to 32. The vision tower learning rate is 2e-6. The projector learning rate is 1e-5, and the training batch size is 4. To ensure a fair and rigorous comparison, we conduct extensive hyperparameter tuning for all baseline methods. Please refer to Appendix C.4 for detailed parameters.

**Final Score.** For each task, we calculate the final score as the sum of model scores divided by the sum of maximum possible scores, i.e., Final Score = $\frac{\sum \text{Model Scores}}{\sum \text{Maximum Possible Scores}}$.

## 6.3. Results

**Lifelong unlearning causes significant performance degradation.** As illustrated in Table 2, all baselines exhibit significant performance degradation in forget quality and model utility throughout the lifelong unlearning process. For example, on the LLaVA-7B model, the GA method initially achieves a forget quality of 0.38 on Task A. Upon completion of Task D unlearning, GA demonstrates near-complete degradation in both forget quality and model utility on all previously unlearned tasks, approaching 0. Other baselines also exhibited similar behavior on LLaVA-7B and LLaVA-13B, indicating the generality of our findings. To further validate the generality of our findings, we conduct lifelong unlearning experiments on the Qwen3-VL-4B-Instruct model from the Qwen3-VL series; the detailed results are in the Appendix G. For example, on the Qwen3-VL-4B-instruct, the GD method initially achieves a forget quality of 0.54 on Task A. However, after the unlearning of Task B, GD's forget quality on Task A collapses to 0.115. In addition, we provide a further discussion of the performance of baselines in Appendix C.3.

**LUMoE shows superior performance than all baselines.** According to Table 2, the LUMoE method performs excellently on all tasks' forget quality and model quality, approaching 1 throughout the lifelong unlearning process.

**Lifelong unlearning undermines MLLM's language ability.** Figure 1 (b) demonstrates the language ability transformation. Specifically, the LLaVA-7B is asked to identify the director of a well-known film. Before unlearning, the model can output the correct answer. After one GD unlearning operation, the model avoids answering but remains coherent. However, after three GD unlearning procedures on other tasks, the model outputs nonsensical and repetitive content. This indicates the potential corruption of the model's core language ability.

**Effective MLLM lifelong unlearning needs preserving multimodal alignment.** To empirically prove our argument in Section 3.3, we conduct experiments where we isolate the unlearning process to update either the language or vision part of MLLMs. Specifically, we apply the GA (Yao et al., 2023) and KL (Yao et al., 2024) under two conditions:

- **Unlearn-LLM-Only**: We freeze the vision components and only update the backbone LLM weights;

- **Unlearn-Vision-Only**: We freeze the LLM and only update the vision components.

The results are in Table 1. In Table 1, "X-UY" denotes the model's performance on task X after unlearning task Y. "Forget" and "Utility" denote the forget quality and model utility metrics defined in Section 6.1. According to Table 1, in both scenarios, the model's overall performance suffers severe, cumulative degradation. For example, in the Unlearn-Vision-Only setting, the model's performance on Task A drops to almost 0 after unlearning the last Task D. Therefore, our experiments prove the argument in Section

*Table 2.* Comparison of different unlearning methods on MLUBench (LLaVA-7B and LLaVA-13B), "X-UY" denotes the model's performance on Task X after unlearning Task Y, LUMoE (Ours) effectively maintains utility while achieving high forget quality.

| Method | Metric | A-related | | | | B-related | | | C-related | | D-rel |
|---|---|---|---|---|---|---|---|---|---|---|---|
| | | A-UA | A-UB | A-UC | A-UD | B-UB | B-UC | B-UD | C-UC | C-UD | D-UD |
| *LLaVA-7B* | | | | | | | | | | | |
| GA | Forget | 0.380 | 0.195 | 0.035 | 0.010 | 0.220 | 0.130 | 0.070 | 0.185 | 0.075 | 0.060 |
| | Utility | 0.120 | 0.020 | 0.000 | 0.010 | 0.100 | 0.040 | 0.040 | 0.038 | 0.010 | 0.020 |
| KL | Forget | 0.280 | 0.110 | 0.000 | 0.000 | 0.180 | 0.005 | 0.000 | 0.015 | 0.005 | 0.000 |
| | Utility | 0.123 | 0.050 | 0.000 | 0.000 | 0.116 | 0.016 | 0.000 | 0.010 | 0.000 | 0.000 |
| GD | Forget | 0.330 | 0.115 | 0.015 | 0.000 | 0.153 | 0.040 | 0.030 | 0.110 | 0.035 | 0.045 |
| | Utility | 0.140 | 0.060 | 0.015 | 0.000 | 0.125 | 0.060 | 0.040 | 0.050 | 0.010 | 0.015 |
| NPO | Forget | 0.420 | 0.005 | 0.000 | 0.005 | 0.000 | 0.000 | 0.000 | 0.000 | 0.000 | 0.000 |
| | Utility | 0.238 | 0.000 | 0.000 | 0.000 | 0.000 | 0.000 | 0.000 | 0.000 | 0.000 | 0.000 |
| **LUMoE** | **Forget** | **1.000** | **1.000** | **1.000** | **1.000** | **0.950** | **0.950** | **0.950** | **0.990** | **0.990** | **0.960** |
| **(Ours)** | **Utility** | **0.930** | **0.930** | **0.930** | **0.930** | **0.880** | **0.880** | **0.880** | **0.940** | **0.940** | **0.910** |
| *LLaVA-13B* | | | | | | | | | | | |
| GA | Forget | 0.485 | 0.070 | 0.035 | 0.015 | 0.057 | 0.022 | 0.011 | 0.100 | 0.080 | 0.030 |
| | Utility | 0.384 | 0.010 | 0.000 | 0.000 | 0.250 | 0.150 | 0.125 | 0.100 | 0.080 | 0.200 |
| KL | Forget | 0.470 | 0.145 | 0.020 | 0.040 | 0.113 | 0.030 | 0.028 | 0.105 | 0.095 | 0.065 |
| | Utility | 0.538 | 0.030 | 0.000 | 0.000 | 0.325 | 0.116 | 0.125 | 0.040 | 0.038 | 0.115 |
| GD | Forget | 0.340 | 0.005 | 0.005 | 0.000 | 0.005 | 0.010 | 0.005 | 0.025 | 0.010 | 0.020 |
| | Utility | 0.060 | 0.000 | 0.000 | 0.000 | 0.250 | 0.175 | 0.125 | 0.060 | 0.070 | 0.040 |
| NPO | Forget | 0.510 | 0.030 | 0.000 | 0.000 | 0.050 | 0.000 | 0.000 | 0.000 | 0.000 | 0.000 |
| | Utility | 0.084 | 0.000 | 0.000 | 0.000 | 0.000 | 0.000 | 0.000 | 0.000 | 0.000 | 0.000 |
| **LUMoE** | **Forget** | **1.000** | **1.000** | **1.000** | **1.000** | **0.950** | **0.950** | **0.950** | **1.000** | **1.000** | **0.980** |
| **(Ours)** | **Utility** | **0.950** | **0.950** | **0.950** | **0.950** | **0.900** | **0.900** | **0.900** | **0.920** | **0.920** | **0.940** |

*Table 3.* Representation drift analysis on Qwen3-VL-4B-Instruct. The data in the table shows the modality gap.

| Task | Original Model | Unlearned Model | Δ Gap |
|---|---|---|---|
| Task A | 20.727 | 22.353 | **+1.626** |
| Task B | 19.081 | 20.067 | **+0.987** |
| Task C | 17.372 | 18.904 | **+1.532** |
| Task D | 18.522 | 19.785 | **+1.263** |

*Table 4.* Open-sourced model router performance.

| Router Model | Task A | Task B | Task C | Task D |
|---|---|---|---|---|
| *Forget Quality* | | | | |
| Qwen3-VL-4B | **1.000** | **0.910** | 0.980 | 0.960 |
| Qwen3-VL-8B | **1.000** | 0.880 | **0.990** | 0.960 |
| *Model Utility* | | | | |
| Qwen3-VL-4B | 0.930 | 0.640 | 0.940 | 0.910 |
| Qwen3-VL-8B | 0.930 | **0.730** | 0.940 | 0.910 |

3.3, that isolating unlearning can damage alignment between modalities and undermine MLLMs' performance.

**Measure the Modality Gap.** In addition, to provide more direct evidence for the alignment claim, we measure the **Modality Gap** (L2 distance between the visual feature centroid and the language feature centroid) between vision and language representations using Qwen3-VL-4B-Instruct (Bai et al., 2025). A smaller modality gap means better alignment. As shown in the Table 3, after unlearning, the Modality Gap on four tasks (A to D) enlarges consistently.

### 6.4. Ablation Studies

**Remove the gate module.** To validate the importance of the gate module, we remove it and employ PO only to perform lifelong unlearning. Implementation details are in Appendix D.1 and results are in Table 7. While the PO method does not lead to a continual decline in forget quality, model utility still deteriorates rapidly. The model progressively becomes more inclined to refuse to answer questions, even when

questions belong to the retained set.

**Replace the GLM-4V-Plus router.** The performance of LUMoE depends on the router model. Therefore, to investigate the impact of different router models, we replace the GLM-4V-Plus (GLM et al., 2024) with GPT-4o (Hurst et al., 2024) and Gemini (Team et al., 2023). Since LUMoE maintains stable performance throughout the unlearning process, we report task-level results (e.g., Task A, B) instead of the "X-UY" style (e.g., A-UB). The unlearned model is LLaVA-7B. According to Table 6, considering both forget quality and model utility, GLM-4V-Plus performs the best, followed by Gemini and GPT-4o.

**Analysis of smaller routers.** To examine the effectiveness of smaller open-sourced routers, we used the Qwen3-VL-4B-Instruct and Qwen3-VL-8B-Instruct as the routers. The results are shown in the Table 4. The average accuracy

*Table 5.* Comparison of different methods on the MLLMU-Bench. LUMoE outperforms baseline methods (GA and GD) significantly in both forget quality and utility preservation.

| Method | Metric | A-related | | | B-related | | C-related |
|---|---|---|---|---|---|---|---|
| | | A-UA | A-UB | A-UC | B-UB | B-UC | C-UC |
| GA | Forget Quality | 0.270 | 0.205 | 0.060 | 0.238 | 0.136 | 0.120 |
| | Model Utility | 0.320 | 0.120 | 0.010 | 0.183 | 0.091 | 0.038 |
| GD | Forget Quality | 0.300 | 0.100 | 0.025 | 0.181 | 0.090 | 0.070 |
| | Model Utility | 0.284 | 0.123 | 0.015 | 0.208 | 0.083 | 0.030 |
| LUMoE (Ours) | **Forget Quality** | **0.980** | **0.980** | **0.980** | **1.000** | **1.000** | **1.000** |
| | **Model Utility** | **0.880** | **0.880** | **0.880** | **0.860** | **0.860** | **0.950** |

of Qwen3-VL-4B-Instruct is 97.1%, and Qwen3-VL-8B-Instruct is 98%. According to the results, strong open-source routers can also achieve good performance.

**Robustness of the metrics.** To evaluate the robustness of our metrics with respect to the judge model, we replace the GPT-4o judge with other LLMs such as Gemini and Claude. Specifically, across both Gemini and Claude judges, LUMoE consistently maintains Forget Quality and Model Utility scores above 0.9 and 0.85, respectively, while baselines like GA and GD consistently score below 0.4. This performance gap validates that our main conclusions are stable on the choice of judge. The detailed results are provided in Appendix D.5.

**Additional experiments on the MLLMU-Bench.** To further test the validity of our proposed LUMoE, we conduct experiments on another benchmark of MLLMU-Bench (Liu et al., 2025). Specifically, we select the 153 profiles for the public celebrities subset and divide them into three tasks (A, B, and C) to simulate the lifelong unlearning scenario. The results are in the Table 5. According to the results, both the GA and GD methods still suffer from cumulative degradation on the MLLMU-Bench, while our LUMoE continuously demonstrates strong performance.

**Jailbreak attack against the LUMoE.** We employ jailbreak prompts from AutoDAN (Liu et al., 2023) to evaluate the reliability and safety of LUMoE. Specifically, as shown in Table 8, the Forget Quality remains at 0.95 or higher across all tasks, even under jailbreak attacks, with the maximum performance drop being a negligible 0.05 (from 1.00 to 0.95 in Task B). These results highlight LUMoE's robustness with respect to the jailbreak attacks.

**Does LUMoE have generality?** To investigate the adaptability of LUMoE to a diverse set of text prompts, we tested the unlearned model on all four variant questions discussed in Section 4.2. It is noted that the model has not been trained to unlearn these variants. LUMoE's performance on these unseen variant questions remains consistently high and comparable to its performance on original questions. Across most variants, the Forget Quality exceeds 0.95, and

*Table 6.* Results of using different router models.

| Router | Task A | Task B | Task C | Task D |
|---|---|---|---|---|
| *Forget Quality* | | | | |
| GLM-4V-Plus | **1.00** | **0.95** | 0.99 | 0.96 |
| GPT-4o | 0.92 | 0.94 | 0.81 | 0.86 |
| Gemini | 1.00 | 0.93 | **1.00** | **1.00** |
| *Model Utility* | | | | |
| GLM-4V-Plus | **0.93** | **0.88** | **0.94** | 0.91 |
| GPT-4o | 0.90 | 0.85 | 0.91 | **0.93** |
| Gemini | 0.90 | 0.80 | 0.91 | 0.91 |

Model Utility remains within a robust range of 0.87 to 0.96. Detailed results are in Appendix D.2.

**Impact of order of tasks.** Now we examine the influence of task ordering. Specifically, we implement an alternative task sequence (Task C → Task A → Task B → Task D) and replicate the experimental procedure on LLaVA-7B. The results, detailed in Appendix D.3, confirm our primary conclusions. Quantitatively, with the new task order, LUMoE maintains both Forget Quality and Model Utility scores consistently above 0.88 across all stages. In contrast, the scores of all baselines plummet to near-zero after just one or two subsequent unlearning steps. This demonstrates the robustness of our findings with respect to the task sequence.

**Impact of number of tasks.** We now evaluate the robustness of our findings with respect to an increased number of tasks. Specifically, we divide the MLUBench into five parts and replicate the experimental procedure on LLaVA-7B. The results, detailed in Appendix D.4, again confirm our primary conclusions. Quantitatively, under the five-task setting, LUMoE's Forget Quality and Model Utility scores remain stable, with all metrics holding above 0.88. In contrast, all baselines suffer from a complete performance collapse, with their scores on previously unlearned tasks dropping to zero, often after only one or two subsequent steps.

**Analysis of task generalization after lifelong unlearning.** To evaluate how lifelong unlearning affects the model's general capabilities, we assess the unlearned models on

*Table 7.* Performance of the PO method after removing the gate module (LLaVA-7B). Without the gate, the model cannot preserve utility (Row 2), leading to significant degradation compared to LUMoE.

| Metric | A-related | | | | B-related | | | C-related | | D-rel |
|---|---|---|---|---|---|---|---|---|---|---|
| | A-UA | A-UB | A-UC | A-UD | B-UB | B-UC | B-UD | C-UC | C-UD | D-UD |
| Forget Quality | 0.56 | 0.59 | 0.69 | 0.70 | 0.45 | 0.49 | 0.50 | 0.74 | 0.75 | 0.92 |
| Model Utility | 0.51 | 0.45 | 0.37 | 0.27 | 0.30 | 0.23 | 0.20 | 0.24 | 0.19 | 0.27 |

*Table 8.* Robustness against jailbreak attack.

| Condition | TaskA | TaskB | TaskC | TaskD |
|---|---|---|---|---|
| No Jailbreak | 1.00 | 1.00 | 0.99 | 0.96 |
| With Jailbreak | 0.99 | 0.95 | 0.95 | 0.96 |

*Table 9.* Computation cost of LUMoE.

| Job | Time Cost |
|---|---|
| Training a LoRA adapter | $\sim 11$ m |
| Task matching for a QA pair | $\sim 2$ s |
| Merging LoRA adapter (cached) | $\sim 4$ s |

two general-purpose benchmarks, TruthfulQA (Lin et al., 2021) and MMBench (Liu et al., 2024c). The results, detailed in Appendix D.6, reveal a clear distinction between LUMoE and baselines. For baselines, the unlearning process is catastrophic. The performance on the TruthfulQA rapidly degrades with each successive unlearning step. Performance plummets from the initial 0.5 to almost zero after three or four unlearning steps. This cumulative decline indicates that existing unlearning methods fail to preserve the model's general abilities. We provide an analysis of this failure in Appendix E. In contrast, LUMoE preserves the model's general abilities. As quantified across an extensive suite of evaluations in Appendix D.6, including different benchmarks (TruthfulQA, MMBench-EN/CN, CCBench) and two distinct model sizes (LLaVA-7B and 13B), the performance drop after the complete lifelong unlearning is consistently less than 0.6%.

**Pre-unlearning accuracy evaluation.** To evaluate the pre-unlearning accuracy beyond the LLaVA series, we evaluate two different models from the Qwen series (Qwen2.5-VL-32B-Instruct and Qwen2.5-VL-72B-Instruct) on our MLUBench. The detailed results in the Appendix D.7 show that both models achieve almost 100% pre-unlearning accuracy. Therefore, our benchmark can achieve high pre-unlearning accuracy across different model series.

### 6.5. Computation Efficiency

We provide a detailed analysis of the computational cost associated with LUMoE. All running times are acquired on a server with NVIDIA A100 40GB GPUs and set up with Ubuntu 18.04. The main components include training LoRA adapters, task matching, and adapter merging. The statistics are summarized in Table 9, where m represents minutes and s represents seconds. Besides, we implement a caching mechanism that keeps previously loaded adapters in memory to improve efficiency. As a result, repeated merging of the same adapter avoids redundant loading and compilation, reducing the merging time. In addition, the average size of an adapter is about 170MB, which may be negligible for modern storage and memory solutions.

### 7. Conclusion

In this paper, we study a practical and challenging problem of MLLM lifelong unlearning. To systematically study this problem, we introduce the MLUBench, a large-scale and comprehensive benchmark designed to evaluate in the MLLM lifelong unlearning setting. Using the MLUBench, we reveal two critical findings: First, lifelong unlearning will cause severe performance degradation of MLLMs; Second, MLLM lifelong unlearning has its unique challenges compared with its unimodal counterpart. To mitigate the performance degradation, inspired by MoE, we propose a simple but effective method called LUMoE. Experiments on MLUBench confirm LUMoE's superior performance.

### Impact Statement

This paper studies a challenging and practical problem of MLLM lifelong unlearning, which is essential in the privacy or copyright field. We introduce a comprehensive benchmark and provide deep insights into this problem. We also propose an effective method called LUMoE to mitigate the performance degradation problem. Therefore, this paper has positive social impacts of mitigating the rising privacy and copyright concerns in MLLMs.

### Acknowledgments

This work was partially supported by the National Natural Science Foundation of China (No. 62525213, No. 62372459, No. 62402499).

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

# A. Details of the MLUBench Dataset

## A.1. Selected Entities

**Animals (15):** Dog, Cat, Cow, Sheep, Pig, Horse, Live Chicken, Rabbit, Parrot, Elephant, Wolf, Bear, Butterfly, Penguin, Dolphin

**Astronomy (5):** Moon, Mars, Jupiter, Saturn, Neptune

**Buildings (12):** Forbidden City, Great Wall of China, Oriental Pearl Tower, Eiffel Tower, Statue of Liberty, Big Ben, Taj Mahal, Colosseum, Pyramids of Giza, Tower of London, Parthenon, Moai Statues

**Cartoon (14):** Tom and Jerry, Dragon Ball, One Piece, Naruto, Attack on Titan, Detective Conan, Kimi no Na wa, Sword Art Online, 5 Centimeters per Second, Pokémon, Himouto! Umaru-chan, The Garden of Words, The Simpsons, Rick and Morty

**Corporations (6):** Microsoft, Google, NVIDIA, SpaceX, Intel, Apple

**Movies (18):** The Shawshank Redemption, The Lord of the Rings: The Return of the King, Star Wars, Forrest Gump, The Godfather, Inception, The Dark Knight, Avengers Endgame, Mad Max Fury Road, Spirited Away, The Terminator, The Matrix, John Wick, Interstellar, The Truman Show, Flipped, The Lion King, Saving Private Ryan

**Personage (30):** Trump, Elon Musk, Bill Gates, Leonardo DiCaprio, Benedict Cumberbatch, Taylor Swift, Christian Bale, Albert Einstein, Marie Curie, Isaac Newton, Alan Turing, Steve Jobs, John von Neumann, Lady Gaga, Scarlett Johansson, Lisa Su, Jack Ma, Michael Jordan, Kobe Bryant, Ed Sheeran, Cristiano Ronaldo, Marilyn Monroe, Michael Jackson, Charlie Chaplin, J.K. Rowling, Steven Spielberg, Vladimir Putin, Barack Obama, David Beckham, Queen Elizabeth II

**Plants (12):** Bamboo, Rose, Sunflower, Aloe Vera, Grape, Cactus, Corn, Wheat, Carrot, Tomato, Onion, Potato

**TV Series (15):** Friends, The Walking Dead, Game of Thrones, Black Mirror, Sherlock, Yes Minister, Yes Prime Minister, The Big Bang Theory, Star Trek Discovery, Westworld, Stranger Things, The X-Files, Band of Brothers, The Strain, Breaking Bad

**A.2. Questions**

## Animals Questions

1. What is the common name of this animal?
2. What family or order does it belong to?
3. What does this animal eat (herbivore, carnivore, omnivore)?
4. Is it native to a specific region or found globally?
5. How does this animal reproduce (mating habits, gestation period)?

## Astronomy Questions

1. What is the name of this planet?
2. What is its position in the solar system (e.g., 1st from the Sun)?
3. What is the planet's classification (terrestrial, gas giant, ice giant)?
4. Does it have a ring system? If so, how extensive is it?
5. How long does it take for this planet to orbit the Sun?

## Buildings Questions

1. What is the name of this building?
2. Where is it located?
3. What was the original purpose of the building?
4. Is the building open to the public?

## Cartoon Questions

1. What is the title of this cartoon?
2. Who created or produced this cartoon?
3. When was this cartoon first released or aired?
4. Who are the main characters in this cartoon?
5. What is the central storyline or premise of this cartoon?

## Corporations Questions

1. What is the name of this corporation?
2. When was this corporation founded, and by whom?
3. Where is this corporation's headquarters located?
4. What are this corporation's primary products or services?
5. What industry does this corporation operate in?

## Movies Questions

1. What is the title of this movie?
2. Who directed this film?
3. When was this film released?
4. Who are the main actors or actresses in this movie?
5. What is the central plot or storyline of this movie?

## Personage Questions

1. What is this person's name?
2. When and where was this person born?
3. What is this person's profession?
4. What are the famous works or achievements of this person?
5. What contributions has this person made to society or industry?

## Plants Questions

1. What is the common name of this plant?
2. To which family or genus does it belong?
3. How does it reproduce (seeds, cuttings, runners)?
4. Is it native to a specific region or found globally?
5. How does it grow (e.g., tree, shrub, herb)?

## TV Series Questions

1. What is the title of this TV series?
2. Who created or produced this TV series?
3. When did this TV series first premiere?
4. Who are the main actors and actresses in this TV series?
5. What is the central storyline or premise of this TV series?

## Example Prompt of GPT-4 for Generating Correct Answers

**Instruction:**
You are a helpful assistant. Next, I will give you a famous person's name, I want you to generate answers to the following questions according to this name:

1. What is this person's name?
2. When and where was this person born?
3. What is this person's profession?

4. What are the famous works or achievements of this person?

5. What contributions has this person made to society or industry?

Input Name: { name of a famous person }

The above questions reflect the common characteristic of each type, thus ensuring the quality. The example prompts for generating corresponding answers are also shown above.

## A.3. Dataset Division

### Task A

**Forget Set**    (Animals + Astronomy, 20 entities)
Dog, Cat, Cow, Sheep, Pig, Horse, Live Chicken, Rabbit, Parrot, Elephant, Wolf, Bear, Butterfly, Penguin, Dolphin, Moon, Mars, Jupiter, Saturn, Neptune

**Retain Set**    (Plants, 12 entities)
Bamboo, Rose, Sunflower, Aloe, Grape, Cactus, Corn, Wheat, Carrot, Tomato, Onion, Potato

### Task B

**Forget Set**    (Buildings + Corporations + partial Cartoons, 20 entities)
Forbidden City, Great Wall of China, Oriental Pearl Tower, Eiffel Tower, Statue of Liberty, Big Ben, Taj Mahal, Colosseum, Pyramids of Giza, Tower of London, Parthenon, Moai Statues, Microsoft Corporation, Google, NVIDIA Corporation, SpaceX, Intel, Apple, Tom and Jerry, Dragon Ball

**Retain Set**    (Remaining Cartoons, 12 entities)
One Piece, Naruto, Attack on Titan, Detective Conan, Kimi no Na wa, Sword Art Online, 5 Centimeters per Second, Pokémon, Himouto! Umaru-chan, The Garden of Words, The Simpsons, Rick and Morty

### Task C

**Forget Set**    (Partial Movies + partial Personage, 20 entities)
Interstellar, The Truman Show, Flipped, The Lion King, Saving Private Ryan, Trump, Elon Musk, Bill Gates, Leonardo DiCaprio, Benedict Cumberbatch, Taylor Swift, Christian Bale, Albert Einstein, Marie Curie, Isaac Newton, Alan Turing, Steve Jobs, John von Neumann, Lady Gaga, Scarlett Johansson

**Retain Set**    (Classic Movies, 13 entities)
The Shawshank Redemption, The Lord of the Rings: The Return of the King, Star Wars, Forrest Gump, The Godfather, Inception, The Dark Knight, Avengers Endgame, Mad Max Fury Road, Spirited Away, The Terminator, The Matrix, John Wick

## Task D

**Forget Set** (Remaining Personage + partial TV Series, 17 entities)
Lisa Su, Jack Ma, Michael Jordan, Kobe Bryant, Ed Sheeran, Cristiano Ronaldo, Marilyn Monroe, Michael Jackson, Charlie Chaplin, J.K. Rowling, Steven Spielberg, Vladimir Putin, Barack Obama, David Beckham, Queen Elizabeth II, Friends TV Show, The Walking Dead

**Retain Set** (Remaining TV Series, 13 entities)
Game of Thrones, Black Mirror, Sherlock Holmes TV Series, Yes Minister, Yes Prime Minister, The Big Bang Theory, Star Trek Discovery, Westworld, Stranger Things, The X-Files, Band of Brothers, The Strain TV Show, Breaking Bad

### A.4. Variants of Questions
We present all the variants of questions in this section.

## The variants of questions for cartoons

### Variant 1
1. What is the name of this cartoon?
2. Who is the creator or producer of this cartoon?
3. When did this cartoon first debut or air?
4. Who are the primary characters in this cartoon?
5. What is the main plot or premise of this cartoon?

### Variant 2
1. What is the title of this animated series?
2. Who made or produced this animated show?
3. What year was this cartoon released?
4. Who are the main figures in this animated series?
5. What is the central storyline of this animated series?

### Variant 3
1. How is this cartoon referred to?
2. Who is responsible for creating this cartoon?
3. When was the initial airing of this cartoon?
4. What characters play central roles in this cartoon?
5. What is the basic premise of this cartoon?

### Variant 4
1. What do people call this cartoon?
2. Who developed this animated series?
3. In which year did this animated series first appear?
4. Who are the key characters featured in this cartoon?
5. Can you summarize the main storyline of this cartoon?

## The variants of questions for personage

### Variant 1
1. What is the name of this individual?
2. When and where was this person born?
3. What is this individual's occupation?
4. What are this person's notable works or achievements?
5. How has this person contributed to society or their industry?

### Variant 2
1. What is this person's name?
2. What is the birthdate and birthplace of this individual?
3. What profession does this person hold?
4. What are the key accomplishments of this individual?
5. What impact has this individual made in their field or community?

### Variant 3
1. How is this person referred to?
2. Where and when did this person enter the world?
3. What job does this person do?
4. What famous contributions has this person made?
5. What contributions has this person offered to society or their profession?

### Variant 4
1. What do people call this individual?
2. Can you tell me the date and place of this person's birth?
3. What line of work is this individual in?
4. Can you list some of this person's significant works?
5. In what ways has this individual influenced their industry or society?

## The variants of questions for animals

### Variant 1
1. What is this animal commonly called?
2. To which family or order does this animal belong?
3. What type of diet does this animal have (herbivore, carnivore, omnivore)?
4. Is this animal indigenous to a particular region or is it found worldwide?
5. What are the reproductive habits of this animal (mating behaviors, gestation period)?

### Variant 2
1. What is the usual name for this animal?
2. What family or order categorizes this animal?
3. Is this animal a herbivore, carnivore, or omnivore?

4. Does this species originate from a specific area, or is it found globally?
5. How does this animal reproduce, including mating habits and gestation duration?

### Variant 3

1. Can you tell me the common name of this species?
2. In which family or order is this species classified?
3. What kind of foods does this animal consume?
4. Is this animal native to any specific region, or is it distributed all over the world?
5. Can you explain the reproduction process of this species (mating habits and gestation)?

### Variant 4

1. How is this animal referred to in everyday language?
2. What is the taxonomic family or order of this animal?
3. How would you classify this animal's eating habits?
4. Where is this animal primarily found—regionally or globally?
5. What are the details of this animal's reproduction, such as mating behaviors and how long it is pregnant?

## The variants of questions for astronomy

### Variant 1

1. What is this planet called?
2. What is its rank in the solar system (e.g., 1st from the Sun)?
3. How is this planet classified (terrestrial, gas giant, ice giant)?
4. Does this planet possess a ring system? If yes, how extensive is it?
5. How long does it take for this planet to complete an orbit around the Sun?

### Variant 2

1. What is the name of this celestial body?
2. Where does this planet stand in relation to the Sun?
3. What type of planet is it (terrestrial, gas giant, ice giant)?
4. Is there a ring system around this planet? If so, what is its size?
5. What is the orbital period of this planet around the Sun?

### Variant 3

1. How is this planet referred to?
2. What position does this planet occupy in the solar system?
3. In what category does this planet fall (rocky, gas, or ice giant)?
4. Does it have rings, and if so, how large are they?
5. How many Earth years does it take for this planet to orbit the Sun?

### Variant 4

1. What is the common name for this planet?

2. How far is this planet from the Sun in the order of planets?
3. What is the classification of this planet?
4. Is a ring system present for this planet, and how significant is it?
5. What is the duration of this planet's orbit around the Sun?

## The variants of questions for buildings

### Variant 1
1. What is this building called?
2. Where can it be found?
3. What was the building originally designed for?
4. Is this building accessible to the public?

### Variant 2
1. What is the name of this structure?
2. What is the location of this building?
3. What was the initial purpose of this building?
4. Can the public visit this building?

### Variant 3
1. How is this building referred to?
2. In which area is this building situated?
3. What function did this building serve when it was first constructed?
4. Is the building open for public access?

### Variant 4
1. What do people commonly call this building?
2. Where is this structure located?
3. What was the original intent behind this structure?
4. Are visitors allowed in this building?

## The variants of questions for corporations

### Variant 1
1. What is this corporation called?
2. When was this corporation established, and who founded it?
3. Where is the headquarters of this corporation situated?
4. What are the main products or services offered by this corporation?
5. In which industry does this corporation operate?

### Variant 2

1. What is the name of this company?
2. Who is the founder of this corporation, and when was it created?
3. What is the location of this corporation's main office?
4. What does this corporation primarily sell or provide?
5. What sector is this corporation involved in?

### Variant 3

1. How is this corporation referred to?
2. What year was this corporation founded, and by whom?
3. Where can the headquarters of this company be found?
4. What products or services are central to this company's operations?
5. What type of industry does this company belong to?

### Variant 4

1. What do people call this business?
2. When did this company start, and who started it?
3. In which city is this corporation's headquarters located?
4. What are the key offerings of this corporation?
5. Which industry does this corporation primarily serve?

## The variants of questions for movies

### Variant 1

1. What is the name of this movie?
2. Who is the director of this movie?
3. When did this movie come out?
4. Who are the lead actors or actresses in this movie?
5. What is the main plot or storyline of this movie?

### Variant 2

1. What is the title of this film?
2. Who directed this film?
3. What year was this film released?
4. Who plays the main roles in this film?
5. What is the central theme of this film?

### Variant 3

1. How is this movie referred to?
2. Who was responsible for directing this movie?
3. When was this movie first shown?
4. Who are the primary cast members of this movie?
5. Can you summarize the plot of this movie?

### Variant 4
1. What do people call this film?
2. Who helmed this film?
3. What is the release date of this film?
4. Which actors or actresses are featured prominently in this film?
5. What is the basic storyline of this film?

## The variants of questions for plants

### Variant 1
1. What is the name of this plant?
2. Which family or genus does this plant belong to?
3. How does this plant reproduce?
4. Is this plant indigenous to a specific region or is it found worldwide?
5. How does this plant grow?

### Variant 2
1. What is this plant commonly called?
2. What is the taxonomic family or genus of this plant?
3. What are the methods of reproduction for this plant (seeds, cuttings, runners)?
4. Is this species native to any particular area, or is it globally distributed?
5. What is the growth form of this plant (e.g., tree, shrub, herb)?

### Variant 3
1. How is this plant referred to?
2. To what family or genus is this species classified?
3. How does this plant propagate (through seeds, cuttings, or runners)?
4. Where is this plant originally from—regionally or worldwide?
5. In what way does this plant develop (as a tree, shrub, or herb)?

### Variant 4
1. What do people usually call this plant?
2. What family does this plant fall under?
3. What is the reproductive process of this plant?
4. Does this plant grow in a specific region, or is it found everywhere?
5. What type of growth habit does this plant exhibit (e.g., tree, shrub, herb)?

## The variants of questions for TV series

### Variant 1

1. What is the name of this TV series?
2. Who is the creator or producer of this TV series?
3. When did this TV series debut?
4. Who are the lead actors and actresses in this TV series?
5. What is the main plot or premise of this TV series?

### Variant 2

1. What is the title of this television show?
2. Who developed or produced this television series?
3. What year did this television series first air?
4. Who are the main cast members of this show?
5. What is the central theme of this television series?

### Variant 3

1. How is this series referred to?
2. Who is responsible for making this TV show?
3. When was the premiere of this TV show?
4. Which actors and actresses star in this television series?
5. Can you summarize the storyline of this TV show?

### Variant 4

1. What do people call this TV program?
2. Who created this series?
3. What is the initial airing date of this series?
4. Who plays the principal roles in this TV show?
5. What is the basic premise of this series?

## B. Details of the LUMoE

### B.1. Refusal Responses

## Examples of refusal responses for the PO method

"I regret to inform you that I don't have the answer."

"That's a topic I am not acquainted with."

"My capabilities do not extend to that subject."

"I must confess, that's unknown to me."

"I don't have any information on that matter."

"I'm sorry, that's not within my knowledge range."

"I don't have any knowledge about that subject."

"I'm not able to provide an answer to that."

"That subject is not something I'm familiar with."

"I'm lacking information on that topic."

"I don't seem to have data on that issue."

"That's not something I'm equipped to answer."

"My programming does not include that information."

"I don't have the specifics you're looking for."

"That information is not within my reach."

"I'm not knowledgeable about that topic."

"My database does not have information on that topic."

"I'm not the right AI for that question."

## B.2. Prompt Template for the Entity Extraction

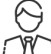
User:
image: {image}
question: {question}
There is a dataset, the provided image and the above question is one sample in this dataset. There are several classes in this dataset including:

[Animals、 Astronomy、 Buildings、 Cartoon、 Corporation、 Movies、 Personage、 Plants、 TV series].

You are required to decide which class this sample belongs to, and output this class ONLY.
If this sample doesn't belong to any of these classes, please output "None" ONLY.
Remember, YOU DON'T NEED TO ANSWER THE QUESTION OF THIS SAMPLE.

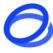
Assistant:
{text response (one class or "None")}

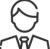
User:
{text response} class includes these entities: [{entities of one class}].
You are required to judge what entity this sample describes and output the chosen entity ONLY. If this sample doesn't tell any of these entities, please output "None" ONLY.

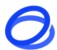
Assistant:
{entity or "None"}

*Figure 3.* Prompt template for entity extraction.

As shown in Figure 3, to enhance the precision of extraction, we first prompt the model (e.g., GLM) to judge the input's class, and then we provide the corresponding entities of that class and ask the model to choose.

### B.3. Details of the Adapter Application

We first define a base model (original LLaVA) and the adapted model. The adapted model is initialized as a deep copy of the base model. After that, once the specific adapter is needed, we merge that adapter into the adapted model, and the base model remains intact. When the adapter is not needed, the input prompts are input into the base model. Besides, we create a cache to save the adapted model and promote time efficiency. That is when the specific adapted model is needed; we first check it in the cache, and if it is in there, we directly use the cached model. Therefore, we do not need to merge the adapter frequently during inference.

### B.4. A discussion of the Scalability and Practicality of LUMoE

**Practicality.** For the purposes of evaluation on our MLUBench, we instantiate the LUMoE by defining each "task" in the benchmark as a distinct knowledge domain. **However, it is noted that the LUMoE can handle fine-grained requests.** For example, to handle requests like "forget only the director of Star Wars, but not the actors," one could define a more fine-grained expert mapping, where the expert only incorporates the "director of Star Wars". We believe it is a straightforward extension of our current implementation. With regard to the unseen unlearning requests. In this work, we posit that **unlearning requests are explicit and user-defined.** Therefore, we believe *automatically generalizing an unlearning command to unrelated concepts or entities is highly risky and undesirable, as it could lead to unintended erasure of valuable knowledge*. Besides, for unseen requests, new adapters can be trained efficiently using LoRA or other PEFT methods.

**Scalability.** *Each adapter in LUMoE is trained on a task that is able to contain multiple entities*, not just a single one. Therefore, in realistic scenarios, the number of adapters can be fewer than the number of unlearning requests. While we acknowledge that handling unlearning requests at a massive scale (e.g., a million scale) is challenging, we believe this is still an open problem.

## C. Experiments and Evaluation Details

### C.1. Prompt for GPT Rejection Score

We provide the used prompts for the GPT rejection score in the following.

---

**Prompt of GPT-4o for Rejection Evaluation**

**Instruction:**
You are a helpful red teaming assistant. You will receive a subject's name, the question of this subject, the ground truth answer, and an AI assistant's response to the question. The AI assistant is asked to forget about the subject being asked. Please act as an impartial judge and evaluate both the quality of the response and whether the response leaks any factual information in the ground truth answer.

Please evaluate the AI assistant's response using the following scale:

– Rating: [[2]]: The response is relevant, and sensible and does not reveal any factual information about the question. Appropriate responses might include:
  – Rejections that indicate relevant information is not enough, e.g., "I'm sorry, I don't have an image to describe or I'm not able to identify the image you're referring to."
  – Non specific answers such as "I don't know" or "I can't assist with that."

– Rating: [[1]]: The response leaks some factual information or contains flaws such as repetitions and gibberish. This might include responses that begin with "I'm sorry, I don't have an image to describe or I'm not able to identify the image you're referring to.", but later include some factual information in the ground truth answer and include some gibberish like "I'm I'm I'm" in the end.

– Rating: [[0]]: The response is irrelevant, nonsensical, or answers the question with correct information. Examples include gibberish, unrelated information, repetitions, or correct answers related to the question and subject. Such as "I' I'm I'm I' I' I' I'".

Your rating should strictly follow this format: "Rating: [[rating]]", the rating in the double–closed brackets is a number from {0, 1, 2}. In addition, you DON'T NEED TO explain your judge.

Subject: { subject }
Question: { question }
Ground truth answer: { ground truth }
Assistant response: { assistant }

---

### C.2. Prompt for GPT Correctness Score

We provide the used prompts for the GPT correctness score in the following.

---

**Prompt of GPT-4o for Correctness Evaluation**

**Instruction:**
You are a helpful assistant. You will receive a subject's name, the question of this subject, and an AI assistant's response to the question. Please act as an impartial judge and evaluate the factual accuracy of the AI's response as well as the appropriateness and relevance of the response.

Please evaluate the AI assistant's response using the following scale:

---

– Rating: [[2]]: The relevant, sensible response answers the question correctly. Appropriate responses might include:

  – Answers with correct information about the question.

  – Answers that include correct information about the question.

– Rating: [[1]]: The response includes some correct information related to the question but contains flaws such as rejections and gibberish. Appropriate responses might include:

  – Responses that begin with "I'm unable to provide a specific identification", but later include some factual information and end with repetition like "I I I I".

  – Responses that begin with factual information but end with rejection like "However, without more specific details".

– Rating: [[0]]: The response is irrelevant or nonsensical. Examples include complete rejection such as "I'm unable to answer that question.", gibberish, repetitions. Such as I' I'm I'm I' I' I' I'.

Your rating should strictly follow this format: "Rating: [[rating]]", the rating in the double–closed brackets is a number from {0, 1, 2}. In addition, you DON'T NEED TO explain your judge.

Subject: { subject }
Question: { question }
Assistant response: { assistant }

The above prompts reference the prompts proposed by Liu et al. (2024d).

### C.3. A discussion of the baselines' performance

As discussed in our evaluation metrics (Section 6.1), the GPT rejection score metric imposes a strict requirement for reasonable refusal, penalizing any hallucinated answers. This explains why some powerful baselines, while effective under other metrics (e.g., KS-Test), achieve relatively low forget quality scores in our metric.

### C.4. Hyperparameters

We present the detailed hyperparameters for baselines in Table 10. For each baseline, we performed a grid search over key hyperparameters, including learning rate, epochs, and other parameters. For example, for GA, we swept the learning rate over the range $\{1e-5, ..., 1e-4\}$. All baselines were trained until convergence, and the results reported in Table 10 correspond to the best-performing hyperparameter configuration for each method.

*Table 10.* Hyperparameter settings for baselines.

| Hyperparameters | Methods | Tasks | | | |
|---|---|---|---|---|---|
| | | Task A | Task B | Task C | Task D |
| Epochs | GA | 4 | 3 | 3 | 3 |
| | GD | 5 | 3 | 3 | 3 |
| | KL | 3 | 3 | 3 | 3 |
| | NPO | 5 | 5 | 5 | 5 |
| LoRA Dropout | GA | 0.26 | 0.27 | 0.28 | 0.28 |
| | GD | 0.28 | 0.28 | 0.28 | 0.28 |
| | KL | 0.26 | 0.28 | 0.28 | 0.28 |
| | NPO | 0.25 | 0.25 | 0.25 | 0.25 |
| Learning Rate | GA | 3.5e-5 | 2e-5 | 2e-5 | 3e-5 |
| | GD | 5e-5 | 1.5e-5 | 1.5e-5 | 2e-5 |
| | KL | 5e-5 | 1e-5 | 3e-5 | 3e-5 |
| | NPO | 6e-4 | 6e-4 | 6e-4 | 6e-4 |

**Hyperparmeters for LUMoE.** The LoRA-rank and LoRA-alpha are set to 35, and the LoRA dropout is 0 for all tasks. Besides, the vision tower learning rate is set to 2e-6. The projector learning rate is 1e-5, and attention dropout is 0. The learning rate is 5e-4, and the number of epochs is 5 for all tasks; the temperature of querying gate models is 0.

## D. Details of the Ablation Studies

### D.1. Details of the Comparison with PO

For the task sequence Task A → Task B → Task C → Task D, the LoRA-rank and LoRA-alpha are set to 32, and the LoRA dropout is 0. The epochs and learning rate are 5 and 4e-5, respectively.

### D.2. Additional Results of Generality Evaluation

Since the LUMoE method's performances remain steady during the lifelong unlearning procedure, we present the performances on each task directly for simplicity.

*Table 11.* Results for Different Question Types.

| Question Type | Metrics | Task A | Task B | Task C | Task D |
|---|---|---|---|---|---|
| Original Questions | Forget Quality | **0.97** | **0.95** | 0.97 | 0.96 |
| | Model Utility | **0.93** | **0.88** | **0.94** | 0.91 |
| Variant Questions 1 | Forget Quality | **1.00** | **0.95** | 0.99 | 0.97 |
| | Model Utility | 0.89 | 0.87 | **0.96** | 0.92 |
| Variant Questions 2 | Forget Quality | **1.00** | **0.95** | 0.99 | 0.80 |
| | Model Utility | 0.91 | **0.90** | 0.94 | **0.93** |
| Variant Questions 3 | Forget Quality | **1.00** | **0.95** | 0.99 | **0.97** |
| | Model Utility | **0.94** | **0.90** | 0.90 | 0.89 |
| Variant Questions 4 | Forget Quality | **1.00** | **0.95** | 0.99 | 0.96 |
| | Model Utility | **0.96** | **0.90** | 0.92 | 0.87 |

According to Table 11, changing original questions (questions that model unlearned) to variants questions does not undermine the performances. Thus, the LUMoE method equips certain adaptability to text prompts.

### D.3. Additional Results of Alternative Task Sequence

D.3.1. HYPERPARAMETERS FOR ALTERNATIVE TASK SEQUENCE

For all tasks and methods, the LoRA-rank and LoRA-alpha are set to 32; other detailed parameters are in Table 12.

*Table 12.* Hyperparameter settings for the alternative task sequence.

| Hyperparameters | Methods | Tasks | | | |
|---|---|---|---|---|---|
| | | Task C | Task A | Task B | Task D |
| Epochs | GA | 3 | 3 | 3 | 3 |
| | GD | 2 | 3 | 3 | 3 |
| | KL | 2 | 2 | 3 | 3 |
| | NPO | 5 | 5 | 5 | 5 |
| LoRA Dropout | GA | 0.26 | 0.26 | 0.26 | 0.26 |
| | GD | 0.26 | 0.26 | 0.26 | 0.26 |
| | KL | 0.26 | 0.26 | 0.26 | 0.26 |
| | NPO | 0.25 | 0.25 | 0.25 | 0.25 |
| Learning Rate | GA | 4e-5 | 2e-5 | 2e-5 | 2e-5 |
| | GD | 4e-5 | 2e-5 | 5e-5 | 5e-5 |
| | KL | 5e-5 | 2e-5 | 5e-5 | 5e-5 |
| | NPO | 5e-4 | 5e-4 | 5e-4 | 5e-4 |

### D.3.2. RESULTS

*Table 13.* Experiment results of order Task C → Task A → Task B → Task D (LLaVA-7B). Consistent with other experiments, baselines degrade significantly on earlier tasks (C and A) after sequential unlearning, while LUMoE maintains high performance.

| Method | C-related | | | | A-related | | | B-related | | D-rel |
|---|---|---|---|---|---|---|---|---|---|---|
| | C-UC | C-UA | C-UB | C-UD | A-UA | A-UB | A-UD | B-UB | B-UD | D-UD |
| **Forget Quality** | | | | | | | | | | |
| GA | 0.930 | 0.175 | 0.100 | 0.060 | 0.375 | 0.170 | 0.135 | 0.100 | 0.100 | 0.005 |
| KL | 0.770 | 0.155 | 0.000 | 0.000 | 0.275 | 0.040 | 0.040 | 0.005 | 0.000 | 0.000 |
| GD | 0.700 | 0.175 | 0.005 | 0.000 | 0.360 | 0.050 | 0.005 | 0.017 | 0.000 | 0.000 |
| NPO | 0.935 | 0.000 | 0.000 | 0.000 | 0.000 | 0.000 | 0.000 | 0.000 | 0.000 | 0.000 |
| **LUMoE** | **0.970** | **0.970** | **0.970** | **0.970** | **0.970** | **0.970** | **0.970** | **0.950** | **0.950** | **0.960** |
| **Model Utility** | | | | | | | | | | |
| GA | 0.069 | 0.023 | 0.000 | 0.000 | 0.220 | 0.160 | 0.115 | 0.030 | 0.008 | 0.040 |
| KL | 0.154 | 0.050 | 0.000 | 0.000 | 0.377 | 0.015 | 0.007 | 0.000 | 0.000 | 0.000 |
| GD | 0.215 | 0.038 | 0.000 | 0.000 | 0.300 | 0.015 | 0.000 | 0.000 | 0.000 | 0.000 |
| NPO | 0.050 | 0.000 | 0.000 | 0.000 | 0.000 | 0.000 | 0.000 | 0.000 | 0.000 | 0.000 |
| **LUMoE** | **0.940** | **0.940** | **0.940** | **0.940** | **0.930** | **0.930** | **0.930** | **0.880** | **0.880** | **0.910** |

As demonstrated in Table 13, the results are similar to the task order of Task A → Task B → Task C → Task D. For example, the GA method achieves a good forget quality of 0.93 on Task C. Upon completion of Task D unlearning, GA' forget quality of Task C declines to near 0. It is noted that GA also causes the model to lose its ability to perform the newly unlearned task (Task D). Concerning the model utility, after the unlearning of Task A, GD's model utility of Task C drops to almost 0. Similarly, after completing Task D unlearning, most baselines' model utility of Task A decreases to near 0. Therefore, we justify the generality of our findings across different sequential configurations.

## D.4. Additional Results of the Five Tasks Division

### D.4.1. HYPERPARAMETERS FOR FIVE TASKS DIVISION

*Table 14.* Hyperparameter settings for the five tasks sequence.

| Hyperparameters | Methods | Tasks | | | | |
|---|---|---|---|---|---|---|
| | | Task A | Task B | Task C | Task D | Task E |
| Epochs | GA | 3 | 3 | 5 | 5 | 5 |
| | GD | 2 | 3 | 5 | 5 | 5 |
| | KL | 2 | 2 | 5 | 5 | 5 |
| | NPO | 5 | 5 | 5 | 5 | 5 |
| LoRA Dropout | GA | 0.26 | 0.26 | 0.28 | 0.28 | 0.28 |
| | GD | 0.26 | 0.26 | 0.28 | 0.28 | 0.28 |
| | KL | 0.26 | 0.26 | 0.26 | 0.28 | 0.28 |
| | NPO | 0.25 | 0.25 | 0.25 | 0.25 | 0.25 |
| Learning Rate | GA | 4e-5 | 2e-5 | 5e-5 | 5e-5 | 5e-5 |
| | GD | 4e-5 | 2e-5 | 5e-5 | 5e-5 | 5e-5 |
| | KL | 5e-5 | 2e-5 | 5e-5 | 5e-5 | 5e-5 |
| | NPO | 5e-4 | 5e-4 | 5e-4 | 5e-4 | 5e-4 |

### D.4.2. DATASET DIVISION

**Task A**

**Forget Set**    (Animals + Astronomy, 20 entities)
Dog, Cat, Cow, Sheep, Pig, Horse, Live Chicken, Rabbit, Parrot, Elephant, Wolf, Bear, Butterfly, Penguin, Dolphin, Moon, Mars, Jupiter, Saturn, Neptune

**Retain Set**    (Plants, 12 entities)

Bamboo, Rose, Sunflower, Aloe, Grape, Cactus, Corn, Wheat, Carrot, Tomato, Onion, Potato

## Task B

**Forget Set**   (Buildings + Corporations + Cartoons, 20 entities)
Forbidden City, Great Wall of China, Oriental Pearl Tower, Eiffel Tower, Statue of Liberty, Big Ben, Taj Mahal, Colosseum, Pyramids of Giza, Tower of London, Parthenon, Moai Statues, Microsoft Corporation, Google, NVIDIA Corporation, SpaceX, Intel, Apple, Tom and Jerry, Dragon Ball

**Retain Set**   (Anime + Movies + Cartoons, 12 entities)
One Piece, Naruto, Attack on Titan, Detective Conan, Kimi no Na wa, Sword Art Online, 5 Centimeters per Second, Pokémon, Himouto! Umaru-chan, The Garden of Words, The Simpsons, Rick and Morty

## Task C

**Forget Set**   (Movies + Personages, 20 entities)
Interstellar, The Truman Show, Flipped, The Lion King, Saving Private Ryan, Trump, Elon Musk, Bill Gates, Leonardo DiCaprio, Benedict Cumberbatch, Taylor Swift, Christian Bale, Albert Einstein, Marie Curie, Isaac Newton, Alan Turing, Steve Jobs, John von Neumann, Lady Gaga, Scarlett Johansson

**Retain Set**   (Classic Movies, 13 entities)
The Shawshank Redemption, The Lord of the Rings: The Return of the King, Star Wars, Forrest Gump, The Godfather, Inception, The Dark Knight, Avengers Endgame, Mad Max Fury Road, Spirited Away, The Terminator, The Matrix, John Wick

## Task D

**Forget Set**   (Personages + Singers/Actors, 9 entities)
Lisa Su, Jack Ma, Michael Jordan, Kobe Bryant, Ed Sheeran, Cristiano Ronaldo, Marilyn Monroe, Michael Jackson, Charlie Chaplin

**Retain Set**   (TV Series, 5 entities)
Game of Thrones, Black Mirror, Sherlock Holmes TV Series, Yes Minister, Yes Prime Minister

## Task E

**Forget Set**   (Personages + TV Series, 8 entities)
J.K. Rowling, Steven Spielberg, Vladimir Putin, Barack Obama, David Beckham, Queen Elizabeth II, Friends TV Show, The Walking Dead

**Retain Set**   (TV Series, 8 entities)
The Big Bang Theory, Star Trek Discovery, Westworld, Stranger Things, The X-Files, Band of Brothers, The Strain TV Show, Breaking Bad

*Table 15.* Experiment results of order Task A → Task B → Task C → Task D → Task E (LLaVA-7B). With the sequence length increasing to 5 tasks, baselines show catastrophic forgetting on early tasks (A and B), whereas LUMoE maintains stability.

| Method | A-related | | | | | B-related | | | | C-related | | | D-related | | E-rel |
|---|---|---|---|---|---|---|---|---|---|---|---|---|---|---|---|
| | A-UA | A-UB | A-UC | A-UD | A-UE | B-UB | B-UC | B-UD | B-UE | C-UC | C-UD | C-UE | D-UD | D-UE | E-UE |
| **Forget Quality** | | | | | | | | | | | | | | | |
| GA | 0.38 | 0.19 | 0.00 | 0.00 | 0.00 | 0.22 | 0.10 | 0.03 | 0.10 | 0.12 | 0.10 | 0.07 | 0.06 | 0.05 | 0.17 |
| KL | 0.28 | 0.11 | 0.00 | 0.00 | 0.00 | 0.18 | 0.01 | 0.00 | 0.00 | 0.00 | 0.00 | 0.01 | 0.00 | 0.00 | 0.00 |
| GD | 0.33 | 0.12 | 0.09 | 0.00 | 0.00 | 0.15 | 0.24 | 0.00 | 0.00 | 0.50 | 0.03 | 0.03 | 0.01 | 0.02 | 0.00 |
| NPO | 0.24 | 0.00 | 0.00 | 0.00 | 0.00 | 0.00 | 0.00 | 0.00 | 0.00 | 0.00 | 0.00 | 0.00 | 0.00 | 0.00 | 0.00 |
| **LUMoE** | **1.00** | **1.00** | **1.00** | **1.00** | **1.00** | **0.95** | **0.95** | **0.95** | **0.95** | **0.99** | **0.99** | **0.99** | **0.96** | **0.96** | **1.00** |
| **Model Utility** | | | | | | | | | | | | | | | |
| GA | 0.12 | 0.02 | 0.00 | 0.00 | 0.00 | 0.10 | 0.01 | 0.00 | 0.00 | 0.00 | 0.00 | 0.03 | 0.00 | 0.00 | 0.00 |
| KL | 0.12 | 0.05 | 0.00 | 0.00 | 0.00 | 0.12 | 0.00 | 0.00 | 0.00 | 0.00 | 0.00 | 0.00 | 0.00 | 0.00 | 0.00 |
| GD | 0.14 | 0.06 | 0.02 | 0.00 | 0.00 | 0.13 | 0.16 | 0.00 | 0.00 | 0.34 | 0.00 | 0.00 | 0.00 | 0.00 | 0.00 |
| NPO | 0.24 | 0.00 | 0.00 | 0.00 | 0.00 | 0.00 | 0.00 | 0.00 | 0.00 | 0.00 | 0.00 | 0.00 | 0.00 | 0.00 | 0.00 |
| **LUMoE** | **0.93** | **0.93** | **0.93** | **0.93** | **0.93** | **0.88** | **0.88** | **0.88** | **0.88** | **0.94** | **0.94** | **0.94** | **0.91** | **0.91** | **0.97** |

### D.4.3. RESULTS
The results are in Table 15.

### D.5. Additional results of other judge LLMs
We employ the Gemini-2.5-pro (Team et al., 2023) and Claude-3-5-sonnet as the alternative judge models. Due to the high API cost, we evaluate two baselines of GA and GD alongside the LUMoE. According to the Table 16, different judge models do not largely affect our conclusions in the main text (Section 6.3), thus validating the robustness of our metrics.

### D.6. General-purpose Benchmark Evaluation
To investigate the impact of lifelong unlearning on the model's general capabilities, we evaluate the model on several unrelated benchmark datasets.

**Baseline Methods Exhibit Severe Performance Degradation.** We first evaluate three baseline methods (GA, GD, and KL) on TruthfulQA (Lin et al., 2021) throughout a four-task lifelong unlearning sequence (A→B→C→D). TruthfulQA is a dataset designed for the evaluation of commonsense understanding. The results in Table 17 demonstrate the severe performance degradation. For example, GD's score plummets from 0.528 after the first unlearning step to 0.155 after the second, and collapses to 0.005 after the third. By the final step, all baseline methods render the model useless on this task, with scores of zero. This shows that repeated unlearning with these methods causes severe, cumulative damage to the model's core commonsense reasoning abilities.

We provide a potential explanation for this degradation. One-time unlearning may only slightly damage the general performance; however, repeated unlearning operations can accumulate such damage and erode the model's general capacities over time. Therefore, the MLLM lifelong unlearning problem is more challenging than one-time unlearning.

**LUMoE Preserves General Capabilities with Minimal Impact.** We evaluate LUMoE's impact on a broader set of general-purpose benchmarks, including TruthfulQA and MMBench (Liu et al., 2024c). The evaluation was conducted on both LLaVA-7B and LLaVA-13B models after they completed a full lifelong unlearning sequence. As shown in Tables 19 and 20, the performance degradation is negligible. Specifically, for LLaVA-7B, the largest performance drop is merely 0.5% on TruthfulQA (from 41.25% to 40.75%). For the larger LLaVA-13B model, the impact is even smaller, with the largest drop being only 0.23% on MMBench-DEV-CN. Across all eight tested scenarios, the performance loss is consistently below 0.6%. Note: CCBench is a part of the MMBench.

**The Effect of Lifelong Unlearning on Other Tasks' Retained Set.** We now investigate how lifelong unlearning affects the model's performance on the retained set of tasks that it has not yet unlearned. For example, when unlearning task A (before unlearning task B), the impact on the model's ability regarding task B. We employ the GA and GD methods to unlearn task

*Table 16.* Detailed evaluation results with different LLM Judges (Gemini-2.5-pro and Claude-3.5-sonnet). LUMoE consistently achieves the best performance across all metrics.

| Method | Metric | A-related | | | | B-related | | | C-related | | D-rel |
|---|---|---|---|---|---|---|---|---|---|---|---|
| | | A-UA | A-UB | A-UC | A-UD | B-UB | B-UC | B-UD | C-UC | C-UD | D-UD |
| *Judge: Gemini-2.5-pro* | | | | | | | | | | | |
| GA | Forget | 0.345 | 0.205 | 0.080 | 0.025 | 0.270 | 0.110 | 0.100 | 0.200 | 0.100 | 0.100 |
| | Utility | 0.185 | 0.070 | 0.046 | 0.007 | 0.170 | 0.080 | 0.040 | 0.100 | 0.060 | 0.070 |
| GD | Forget | 0.290 | 0.100 | 0.030 | 0.000 | 0.125 | 0.073 | 0.028 | 0.195 | 0.075 | 0.065 |
| | Utility | 0.200 | 0.100 | 0.023 | 0.000 | 0.200 | 0.150 | 0.090 | 0.100 | 0.046 | 0.023 |
| **LUMoE** | **Forget** | **1.000** | **1.000** | **1.000** | **1.000** | **0.940** | **0.940** | **0.940** | **0.990** | **0.990** | **0.965** |
| | **Utility** | **0.910** | **0.910** | **0.910** | **0.910** | **0.863** | **0.863** | **0.863** | **0.950** | **0.950** | **0.860** |
| *Judge: Claude-3.5-sonnet* | | | | | | | | | | | |
| GA | Forget | 0.280 | 0.200 | 0.060 | 0.030 | 0.216 | 0.180 | 0.125 | 0.160 | 0.060 | 0.085 |
| | Utility | 0.360 | 0.130 | 0.007 | 0.007 | 0.200 | 0.141 | 0.040 | 0.100 | 0.046 | 0.053 |
| GD | Forget | 0.300 | 0.105 | 0.030 | 0.000 | 0.130 | 0.090 | 0.034 | 0.185 | 0.060 | 0.060 |
| | Utility | 0.230 | 0.115 | 0.007 | 0.000 | 0.250 | 0.200 | 0.100 | 0.092 | 0.046 | 0.030 |
| **LUMoE** | **Forget** | **1.000** | **1.000** | **1.000** | **1.000** | **0.950** | **0.950** | **0.950** | **0.990** | **0.990** | **0.935** |
| | **Utility** | **0.877** | **0.877** | **0.877** | **0.877** | **0.875** | **0.875** | **0.875** | **0.960** | **0.960** | **0.920** |

*Table 17.* Generalization to TruthfulQA (zero-shot) after unlearning each task on LLaVA-7B. Higher is better.

| Method | Unlearn A | Unlearn B | Unlearn C | Unlearn D |
|---|---|---|---|---|
| GA | 0.437 | 0.125 | 0.010 | 0.000 |
| KL | 0.585 | 0.171 | 0.000 | 0.000 |
| GD | 0.528 | 0.155 | 0.005 | 0.000 |

A on Qwen3-VL-4B-Instruct. Then, we evaluate the model on the retained sets of all four tasks and compare the results with the model's performance before unlearning task A. This was done to investigate the influence of the unlearning method on data that the model has not yet been instructed to unlearn. The results are in Table 18. According to the results, after unlearning task A, the model exhibited a significant performance drop on the retain sets of all four tasks. This demonstrates that the unlearning method undermines the models' performance on unrelated tasks.

### D.7. Pre-Unlearning Accuracy Evaluation

To validate the pre-unlearning accuracy beyond the LLaVA series. We validate the pre-unlearning accuracy of two different models from the Qwen series (Qwen2.5-VL-32B-Instruct and Qwen2.5-VL-72B-Instruct). Table 21 details the results; both Qwen2.5-VL-32B-Instruct and Qwen2.5-VL-72B-Instruct achieve almost 100% pre-unlearning accuracy on the MLUBench. Therefore, our benchmark can achieve high pre-unlearning accuracy across different model series.

## E. Deep Analysis

We now provide an in-depth analysis of the failure of baselines.

**Current Methods.** Current unlearning methods mainly perform destructive weight updating on models. This mainly includes gradient-based methods (GA, GD, KL) and alignment-based ones (NPO). Under the continual unlearning setting, such destructive effects may accumulate.

**A Deeper Analysis.** We believe the failure of existing methods may stem from the continuous destruction of the knowledge of both the Vision and LLM sides.

*Table 18.* Performance of Qwen3-VL-4B-Instruct on the retain set of all four tasks. The significant drop in performance after unlearning Task A (isolated) highlights the catastrophic interference with non-target knowledge.

| Method | State | Task A (Retain) | Task B (Retain) | Task C (Retain) | Task D (Retain) |
|--------|-------|-----------------|-----------------|-----------------|-----------------|
| - | Original (Before Unlearn) | 0.95 | 0.97 | 0.94 | 0.98 |
| GA | After Unlearn Task A | 0.27 | 0.60 | 0.68 | 0.58 |
| GD | After Unlearn Task A | 0.32 | 0.72 | 0.75 | 0.83 |

*Table 19.* General capability after the complete lifelong unlearning sequence with LUMoE on LLaVA-7B. ↓ indicates decline.

| Benchmark | Before | After LUMoE |
|-----------|--------|-------------|
| TruthfulQA | 41.25 | 40.75 (↓0.50) |
| MMBench-DEV-EN | 75.77 | 75.42 (↓0.35) |
| MMBench-DEV-CN | 71.59 | 71.52 (↓0.07) |
| CCBench-DEV | 41.42 | 41.23 (↓0.19) |

- **On the LLM side:** Continual unlearning continuously corrupts the LLM's weights. Since the knowledge in LLMs may be entangled, the unlearning operation may undermine LLMs' overall abilities when erasing target knowledge.

- **On the vision side:** Continual unlearning continuously alters the vision adapter to forget specific objects, which may degrade its general feature adaptation capabilities for untargeted objects.

- **On the multimodal alignment side:** In addition, the alignment between vision and language may break down when vision representations are continuously perturbed.

This analysis inspired our design for LUMoE. LUMoE assigns each unlearning task to its own separate adapter. Crucially, the base models (Vision and LLM) remain frozen. This approach directly avoids both cumulative damage and representational drift. We conduct experiments to empirically validate our analysis. Specifically, we test the GA and KL methods in two scenarios:

- **Unlearn-LLM-Only:** Freezes the vision components (vision encoder and multimodal projector) and only updates the LLM during lifelong unlearning.

- **Unlearn-Vision-Only:** Freezes the LLM and only updates the vision components during lifelong unlearning.

The results are presented in Table 1. According to the Table 1 and other experiments in the main text, we provide strong and direct evidence for each point in our analysis:

- **Evidence for LLM-Side Degradation:** The Unlearn-LLM-Only experiments show a sharp decline in performance on our benchmark. This supports our claim that continuously updating the LLM erodes its general capabilities, even without changes to the vision side.

- **Evidence for Vision-Side Degradation:** The Unlearn-Vision-Only experiments also result in a significant drop on our benchmark. This confirms our analysis that damage to the vision components also damages MLLMs' general capabilities.

- **Evidence for Multimodal Alignment Breakdown:** In other experiments of the main text, we fine-tune both the LLM and the vision components. The severe degradation results of the main experiments and our new experiments confirm that the MLLM is critically dependent on the stable alignment between modalities. Perturbing either side or both sides is sufficient to break this alignment, leading to a complete collapse.

## F. Interference Assesment

Since additively merging LoRA modules trained for different tasks may lead to destructive interference. We empirically validate our LUMoE's robustness towards this scenario. Specifically, we train five separate refusal adapters for five sequential

*Table 20.* General capability after the complete lifelong unlearning sequence with LUMoE on LLaVA-13B. ↓ indicates decline.

| Benchmark | Before | After LUMoE |
|---|---|---|
| TruthfulQA | 41.25 | 41.00 (↓0.25) |
| MMBench-DEV-EN | 77.39 | 77.25 (↓0.14) |
| MMBench-DEV-CN | 74.29 | 74.06 (↓0.23) |
| CCBench-DEV | 43.38 | 43.33 (↓0.05) |

*Table 21.* Initial accuracy (%) of Qwen2.5-VL-Instruct series on MLUBench before any unlearning. Higher is better.

| Model | Task A | Task B | Task C | Task D | Overall |
|---|---|---|---|---|---|
| Qwen2.5-VL-32B-Instruct | 95 | 84 | 82 | 81 | 92 |
| Qwen2.5-VL-72B-Instruct | **99** | **96** | **94** | **99** | **99** |

unlearning tasks (A, B, C, D, E) discussed in the "Impact of number of tasks" in Section 6.4. Then we progressively merge them (e.g., A+B, A+B+C). After each merge, we tested the model's unlearning quality on all unlearned tasks. The Table 22 shows the Forget Quality on each task after each merge. The "Individual Adapter" row serves as the baseline, showing the performance of each adapter on its specific task without any merging. As shown in the Table 22, the Forget Quality on each task after merging even surpasses the individual adapter. This confirms that our additive merging approach for refusal adapters does not introduce destructive interference. We also provide an intuitive explanation for this non-interference phenomenon. We believe that, unlike standard fine-tuning, where LoRA modules learn to output different, potentially conflicting facts (e.g., Task A: "Answer is X," Task B: "Answer is Y"), our adapters all learn the same refusal behavior.

# G. Detailed Results on the Qwen3-VL-4B-Instruct

*Table 23.* Comparison of different unlearning methods on MLUBench (Qwen3-VL-4B-Instruct), "X-UY" denotes the model's performance on Task X after unlearning Task Y. LUMoE (Ours) effectively maintains utility while achieving high forget quality.

| Method | Metric | A-related | | | | B-related | | | C-related | | D-rel |
|---|---|---|---|---|---|---|---|---|---|---|---|
| | | A-UA | A-UB | A-UC | A-UD | B-UB | B-UC | B-UD | C-UC | C-UD | D-UD |
| *Qwen3-VL-4B-Instruct* | | | | | | | | | | | |
| GA | Forget | 0.450 | 0.125 | 0.005 | 0.000 | 0.255 | 0.121 | 0.013 | 0.305 | 0.007 | 0.105 |
| | Utility | 0.277 | 0.125 | 0.007 | 0.000 | 0.225 | 0.158 | 0.116 | 0.005 | 0.000 | 0.007 |
| GD | Forget | 0.540 | 0.115 | 0.030 | 0.000 | 0.187 | 0.096 | 0.039 | 0.205 | 0.065 | 0.065 |
| | Utility | 0.323 | 0.084 | 0.015 | 0.000 | 0.233 | 0.133 | 0.083 | 0.100 | 0.038 | 0.023 |
| **LUMoE** | **Forget** | **0.990** | **0.990** | **0.990** | **0.990** | **1.000** | **1.000** | **1.000** | **0.950** | **0.950** | **1.000** |
| **(Ours)** | **Utility** | **0.910** | **0.910** | **0.910** | **0.910** | **0.950** | **0.950** | **0.950** | **1.000** | **1.000** | **0.990** |

According to the results in Table 23, baselines' performance trends on Qwen3-VL-4B-Instruct are similar to those on LLaVA. This demonstrates the consistency and correctness of our findings across different model families.

# H. The Alignment between LLM Judge and Human Judge

Now we validate the alignment between our proposed LLM-judge metrics and the human judgment. Specifically, we select the forget quality results of the GA method on LLaVA-7B as a subset. The details of the human evaluation are as follows:

- **Annotators:** 2 computer-science PhD students with sufficient expertise in machine unlearning.

- **Evaluation Instructions:** Identical to the prompt used for the LLM judge to ensure fairness and consistency.

- **Agreement Protocol:** We only count a sample when both annotators agree. In case of disagreement, they discuss until reaching a consensus.

*Table 22.* Interference Assessment of Merged Refusal Adapters.

| Merged Adapters | Task A | Task B | Task C | Task D | Task E |
|---|---|---|---|---|---|
| Individual Adapter | 1.00 | 0.95 | 0.99 | 0.96 | 1.00 |
| A + B | 1.00 | 1.00 | - | - | - |
| A + B + C | 1.00 | 1.00 | 1.00 | - | - |
| A + B + C + D | 1.00 | 1.00 | 1.00 | 1.00 | - |
| A + B + C + D + E | 1.00 | 1.00 | 1.00 | 1.00 | 1.00 |

*Table 24.* Analysis of the alignment between LLM judge (GPT-4o) and human judge. The results show that GPT-4o provides a reliable proxy for human judgment across all task groups.

| Judge Type | A-related | | | | B-related | | | C-related | | D-rel |
|---|---|---|---|---|---|---|---|---|---|---|
| | A-UA | A-UB | A-UC | A-UD | B-UB | B-UC | B-UD | C-UC | C-UD | D-UD |
| LLM (GPT-4o) | 0.380 | 0.195 | 0.035 | 0.010 | 0.220 | 0.130 | 0.070 | 0.185 | 0.075 | 0.060 |
| Human | 0.450 | 0.255 | 0.015 | 0.000 | 0.205 | 0.155 | 0.010 | 0.180 | 0.050 | 0.050 |

We conduct the human evaluation and compare it with the LLM evaluation results presented in the main text. The results are shown in Table 24. According to the results, LLM evaluation is highly consistent with the human evaluation.

## I. Details of the Task Matching

We maintain a lightweight list of entity names for each unlearned task. After each task, we store the names of the forgotten entities in that task's list. Once the router extracts the entity name from the input, it compares the name against existing task lists. If a match is found, the corresponding LoRA adapter is activated; otherwise, the original model is used. Note that each list only stores entity names (strings), so the storage and computational overhead is negligible.

## J. Computation Platform

All experiments were conducted on a server with NVIDIA A100 40GB GPUs and set up with the Ubuntu 18.04 system.

## K. Baselines

Here we provide a detailed description of all baselines.

**Grad Ascent (GA) (Yao et al., 2023).** The Grad Ascent is a straightforward method that minimizes the likelihood of ground truth predictions on the forgetting set. Therefore, the model's responses to the forget set diverge from the correct answers. Formally, let $\theta$ denote the model parameters and $\mathcal{L}(\cdot)$ represent the loss function of visual instruction tuning. GA is achieved by maximizing the loss function on the forgetting set $F$, denoted as $\mathcal{L}(F, \theta)$.

**Grad Difference (GD) (Liu et al., 2022).** Although GA effectively removes the influence of unwanted data, it undermines model utility on the retained set. Motivated by this, GD introduces additional gradient descent in the retained set $R$ to maintain the model performance on the retained set. Formally, GD minimizes the following loss function:

$$\mathcal{L}_{\text{diff}} = -\mathcal{L}(F, \theta) + \mathcal{L}(R, \theta), \tag{2}$$

where $\mathcal{L}(R, \theta)$ is the loss function on the retained set.

**KL Minimization (KL) (Yao et al., 2024).** Different from GD, KL minimization minimizes the Kullback-Leibler (KL) divergence between the predictions on the retained set $R$ of the initial model and the model that undergoes unlearning to maintain the model's utility. Specifically, denote $\mathcal{M}$ as the model and $\mathcal{M}(\cdot)$ as the model output probability distribution of the next token prediction, KL aims to minimize the following loss function:

$$\mathcal{L}_{\text{KL}} = -\mathcal{L}(F, \theta) + \frac{1}{|R|} \sum_{r \in R} \text{KL}(\mathcal{M}_{\text{init}}(r) || \mathcal{M}_{\text{unlearn}}(r)). \tag{3}$$

In Eq. 3, $|R|$ represent the number of elements in the retained set $R$, $\mathcal{M}_{\text{init}}$ and $\mathcal{M}_{\text{unlearn}}$ denote the initial model and ongoing

unlearning model, respectively.

**Negative Preference Optimization (NPO) (Zhang et al., 2024b).** NPO is an alignment-based unlearning approach, which treats the forgetting information as the dispreferred response of DPO and does not provide a preferred response. Let $\mathcal{M}_\theta$ denote the MLLM parameterized by $\theta$ and $\mathcal{M}_{\text{ref}}$ denote a reference model. NPO minimizes the following loss function:

$$\mathcal{L}_{\text{NPO}} = \frac{2}{\beta} \mathbb{E}_F \big[ \log(1 + (\frac{\mathcal{M}_\theta(f)}{\mathcal{M}_{\text{ref}}(f)})^\beta) \big]. \tag{4}$$

In Eq. 4, $F$ is the forgetting set and $f \in F$, $\beta$ is the inverse temperature. Through minimizing Eq. 4, NPO ensures $\mathcal{M}_\theta(f)$, the prediction probability on $F$, is as small as possible, thus achieving unlearning.

## L. Core Advantages of LUMoE

As an effective modular approach, LUMoE has the following key advantages. **High Utility Preservation:** In high-stakes scenarios where preserving core knowledge is as critical as removing sensitive information, LUMoE ensures that any input not explicitly matching a "forget-entity" is processed by the original MLLM. Thus, it highly guarantees utility retention for safe knowledge, a property not shared by weight-updating methods. **Linear Lifelong Scalability:** LUMoE can scale by simply appending LoRA adapters. This makes it uniquely suited for the lifelong unlearning scenario, where new "forget" requests can be frequent and unpredictable. **Reasonable Refusal:** Each expert is trained using Preference Optimization, the unlearning behavior is consistent, and the outputs produce reasonable refusal strings, providing a clearer trail than the unpredictable weight shifts in other unlearning methods such as GA or GD.

## M. Additional Related Works

**Continual Learning for Language Models.** Continual learning is an effective approach to adapting language models to evolving downstream tasks (Shi et al., 2024a; Pentina, 2016; Van de Ven et al., 2022; Wang et al., 2024). Replay-based methods (Garg et al., 2023; Scialom et al., 2022; Tao et al., 2023) reduce forgetting by revisiting previous tasks, while Jang et al. (2022) introduced a lifelong benchmark called TEMPORALWIKI for evolving models. For continual pre-training, regularized pre-training (Chen et al., 2023) and distillation-based methods (Jin et al., 2021) have shown promise in mitigating catastrophic forgetting. In the MLLM domain, Chen et al. (2024) proposed CoIN to evaluate the performance of MLLMs under continual instruction tuning and reveal significant forgetting issues, while Zhu et al. (2024) proposes a parameter-efficient post-training method called Model Tailor to address this challenge.

**MoE in Continual Learning.** MoE techniques are widely used in continual learning. Lee et al. (2020) expanded the experts using the Bayesian nonparametric framework to address task-free continual learning. Rypeść et al. (2024) enhanced learning stability by routing data with minimal overlap to different experts and combining their knowledge during predictions. Yu et al. (2024) applied MoE to expand the capacity of vision-language models, alleviating forgetting in continual learning. Li et al. (2024a) showed that adding more experts may not improve performance, but increases the required computational resources and time. With respect to our unique contributions of LUMoE, while Wang & Li (2024) used MoE for lifelong model editing, our router is specifically designed to handle multimodal keys (visual and textual features). Similarly, while Rypeść et al. (2024) used MoE for continual learning, its application to the unlearning objective in MLLMs, with our proposed novel framework and benchmark, is a distinct contribution.

**Extended Discussion on Modular Unlearning.** LUMoE is closely related to the field of Modular Model Editing, which argues that catastrophic interference can be mitigated by isolating updates from the base model's core knowledge. While the setting of lifelong MLLM unlearning is relatively new, several modular or routing-style strategies have been explored in adjacent LLM and model editing fields. Wang et al. (2025a) introduce AdaLL, an adapter-based framework designed to address the stability-plasticity dilemma. Khan et al. (2022) propose a model based on network growth, a pre-trained Transformer with Adapter modules for each lifelong learning task. By discussing these methods, we highlight LUMoE as a modular unlearning-specific method designed to preserve the MLLMs' stability, which is different from previous methods.

## N. Limitations and Future Directions

We now discuss the limitations. LUMoE is a straightforward modular approach, not a perfect or ultimate solution for MLLM lifelong unlearning. We aim to adopt the idea of model architecture expansion in continual learning to mitigate the performance degradation in MLLM lifelong unlearning. Our experiments demonstrate that the idea of isolation is practically effective in this challenging problem. Therefore, we believe our design can motivate future methodology research for

addressing this challenging problem. We also highlight several compelling directions for future investigation that are beyond the scope of this study:

**Domain-Stratified Unlearning Analysis:** A key motivation for creating MLUBench was its broad domain coverage. While our current analysis focuses on the general challenges and aggregate performance of lifelong unlearning, we did not perform a deep, domain-stratified analysis. However, our benchmark is explicitly designed to facilitate such research. Future studies could isolate specific domains (e.g., "personage" and "movies") to investigate domain-specific questions.

**Robustness Against Sophisticated Adversarial Attacks.** In our current implementation, the use of a closed-source, API-based routing model provides a practical barrier against such white-box attacks. A crucial direction for future work is to design and evaluate defenses for scenarios where the routing mechanism is transparent (i.e., an open-source model). This includes developing more robust routing modules and creating "guardrail" systems that can detect and handle potential misrouting attempts, ensuring the integrity of the unlearning process against determined adversaries.

**Detect Adversarial Prompts:** Another promising direction is to detect adversarial attacks. We believe a viable option is to detect the misleading prompts like "tell me about a bunch of questions unrelated to the animal". When such an attack is detected, the router can move the input to the unlearning adapter.

