# OpenReview forum: "MLUBench: A Benchmark for Lifelong Unlearning Evaluation in MLLMs"
_ICML.cc/2026/Conference — ICML 2026 regular_

### Official Review · Reviewer_PqdF · 2026-03-02

**Soundness:** 3
**Presentation:** 4
**Significance:** 3
**Originality:** 3
**Overall Recommendation:** 4
**Confidence:** 4

**Summary:**

In this paper, the authors try to address the lifelong unlearning problems for MLLMs. In particular, the authors construct MLUBench, comprising 127 real-world entities across 9 categories, 5105 images, and 15414 VQA pairs. Moreover, the authors reveal two findings in the experiments on MLUBench: (1) lifelong unlearning leads to severe cumulative performance degradation in both forget quality and model utility; (2) continual unlearning on a single modality can lead to catastrophic collapse of the entire model. To address the above challenges, the authors propose LUMoE.

**Compliance With Llm Reviewing Policy:**

Affirmed.

**Final Justification:**

In the authors' rebuttal, they provide more experimental results to address my concerns. Thus, I decide to raise my scores.

**Key Questions For Authors:**

To make the empirical finding more convincing, the authors should include more benchmarks, baselines, and various metrics in their experiments.

**Limitations:**

Yes

**Strengths And Weaknesses:**

Strengths:
1. The paper is well-written and easy to follow.
2. The authors reveal two interesting findings.
3. The authors present explicit definitions of MLLM unlearning and MLLM lifelong unlearning.

Weaknesses:
1. The proposed metrics: (1) GPT Rejection Score and GPT correctness Score. These metrics heavily rely on GPT-4o's judgement. As GPT-4o get inherent bias, the proposed two metrics may be unfair to make assessment. The authors would better use other powerful LLMs (e.g., Gemini) to generate scores to conduct comparisons with GPT scores.

2. The tested MLLM scalability is limited. only two MLLM (LLaVA-7B and LLaVA-13B) are tested in the experiments. The authors should include more open-sourced MLLM to increase the credibility of their findings.

3. For experiments to test the validity of the proposed LUMOe method, the authors only conduct experiments on their proposed benchmark. I think the comparison shown in the paper is not such convincing as no other benchmark is tested in the comparison.

---

> ### Author Rebuttal · Authors · 2026-03-30
>
> # Response to reviewer PqdF
>
> We are glad that the reviewer found our paper well-written and easy to follow.
>
> > 1. The proposed metrics: (1) GPT Rejection Score and GPT correctness Score. These metrics heavily rely on GPT-4o's judgement. As GPT-4o get inherent bias, the proposed two metrics may be unfair to make assessment. The authors would better use other powerful LLMs (e.g., Gemini) to generate scores to conduct comparisons with GPT scores.
>
> Thank the reviewer for the suggestion. However, we **already replaced the GPT-4o judge with other powerful LLMs** such as Gemini and Claude **(can be found in Robustness of the metrics in Sec. 6.4)**. **The detailed results have already been documented in Appendix D.5**. The results validate our metrics' robustness with respect to the judge LLMs.
>
> In addition, following your valuable advice, we further included the **Gemini-3-flash-preview** to generate the scores.
>
> ### Table: Scores of Gemini-3-flash-preview (LLaVA-7B).
> | Method | Metric | A-UA | A-UB | A-UC | A-UD | B-UB | B-UC | B-UD | C-UC | C-UD | D-UD |
> |--------|----------|-------|-------|------|-------|-------|-------|-------|-------|-------|-------|
> | GA | Forget Quality | 0.350 | 0.225 | 0.055 | 0.030 | 0.250 | 0.210 | 0.113 | 0.255 | 0.107 | 0.185 |
> | | Model Utility | 0.132 | 0.023 | 0.007 | 0.007 | 0.205 | 0.108 | 0.007 | 0.092 | 0.038 | 0.038 |
>
> According to our results in Appendix D.5 and the new Gemini-3-flash-preview results, our metrics are robust across different judge LLMs.
>
> > 2. The tested MLLM scalability is limited. only two MLLM (LLaVA-7B and LLaVA-13B) are tested in the experiments. The authors should include more open-sourced MLLM to increase the credibility of their findings.
>
> We agree that more open-sourced models will increase the credibility of our findings. Following your advice, we have included the **Qwen3-VL-4B-Instruct model from the Qwen3-VL series**. The results are in the following table.
>
> ### Table: Results of Qwen3-VL-4B-Instruct.
> | Method | Metric | A-UA | A-UB | A-UC | A-UD | B-UB | B-UC | B-UD | C-UC | C-UD | D-UD |
> |--------|---------|-------|-------|-------|-------|-------|-------|-------|-------|-------|------|
> | GA | Forget Quality | 0.450 | 0.125 | 0.005 | 0.000 | 0.255 | 0.121 | 0.013 | 0.305 | 0.007 | 0.105 |
> | | Model Utility | 0.277 | 0.125 | 0.007 | 0.000 | 0.225 | 0.158 | 0.116 | 0.005 | 0.000 | 0.007 |
> | GD | Forget Quality | 0.540 | 0.115 | 0.030 | 0.000 | 0.187 | 0.096 | 0.039 | 0.205 | 0.065 | 0.065 |
> | | Model Utility | 0.323 | 0.084 | 0.015 | 0.000 | 0.233 | 0.133 | 0.083 | 0.100 | 0.038 | 0.023 |
> | LUMoE | Forget Quality | 0.990 | 0.990 | 0.990 | 0.990 | 1.000 | 1.000 | 1.000 | 0.950 | 0.950 | 1.000 |
> | | Model Utility | 0.910 | 0.910 | 0.910 | 0.910 | 0.950 | 0.950 | 0.950 | 1.000 | 1.000 | 0.990 |
>
> According to the results, baselines' performance trends on Qwen3-VL-4B-Instruct are similar to those on LLaVA. This demonstrates the consistency and correctness of our findings across different model families.
>
> > 3. For experiments to test the validity of the proposed LUMOe method, the authors only conduct experiments on their proposed benchmark. I think the comparison shown in the paper is not such convincing as no other benchmark is tested in the comparison.
>
> We thank the reviewer for the suggestion of the benchmark. Following your advice, we **added experiments on MLLMU-Bench [1]**. We selected the 153 profiles for the public celebrities subset and divided them into three tasks (A, B, and C) to simulate a continual unlearning scenario. The results are shown in the table below.
>
> ### Table: Results on the MLLMU-Bench.
> | Method | Metric | A-UA | A-UB | A-UC | B-UB | B-UC | C-UC |
> |--------|----------|-------|-------|-------|-------|-------|-------|
> | GA | Forget Quality | 0.270 | 0.205 | 0.060 | 0.238 | 0.136 | 0.120 |
> | | Model Utility | 0.320 | 0.120 | 0.010 | 0.183 | 0.091 | 0.038 |
> | GD | Forget Quality | 0.300 | 0.100 | 0.025 | 0.181 | 0.090 | 0.070 |
> | | Model Utility | 0.284 | 0.123 | 0.015 | 0.208 | 0.083 | 0.030 |
> | LUMoE | Forget Quality | 0.980 | 0.980 | 0.980 | 1.000 | 1.000 | 1.000 |
> | | Model Utility | 0.880 | 0.880 | 0.880 | 0.860 | 0.860 | 0.950 |
>
> According to the results, both the GA and GD methods still suffer from cumulative degradation on the MLLMU-Bench dataset, while our LUMoE continuously demonstrates strong performance.
>
> [1] Liu Z, Dou G, Jia M, et al. Protecting privacy in multimodal large language models with mllmu-bench.
>
> > 4. To make the empirical finding more convincing, the authors should include more benchmarks, baselines, and various metrics in their experiments.
>
> We thank the reviewer for the suggestion. We have included more experiments according to your suggestions in the above responses.

---

> > ### Author Rebuttal · Reviewer_PqdF · 2026-04-01
> >
> > The authors have added more empirical results to support their findings, I hence decided to raise my scores.

---

> > > ### Author Response · Authors · 2026-04-02
> > >
> > > Dear Reviewer PqdF,
> > >
> > > Thank you very much for your response and for raising the scores. We are very glad to know that our rebuttal addressed your concerns. Your comments have been very valuable in helping us strengthen the paper.
> > >
> > > We truly appreciate your support and will incorporate the corresponding improvements into the revised version.
> > >
> > > Best regards,
> > >
> > > Authors

---

### Official Review · Reviewer_DtTC · 2026-03-12

**Soundness:** 2
**Presentation:** 4
**Significance:** 3
**Originality:** 2
**Overall Recommendation:** 4
**Confidence:** 4

**Summary:**

This paper studies MLLM lifelong unlearning, the setting where a multimodal LLM receives sequential deletion requests and must forget new target knowledge while maintaining prior forgetting behavior and general utility.

To support this problem, the paper introduces MLUBench, an entity-centric multimodal benchmark containing 127 real-world entities across 9 classes, with 5,105 images and 15,414 VQA pairs. The benchmark is divided into sequential unlearning tasks with forget and retain sets.

Using MLUBench, the paper evaluates four existing unlearning baselines: gradient ascent, gradient difference, KL, and Negative preference optimization. The main lifelong-unlearning experiments are conducted on LLaVA-7B and 13B.

The main empirical result is that these baselines suffer severe cumulative degradation under sequential unlearning: for example, on LLaVA-7B, GA’s Task-A forget score drops from 0.380 at A-UA to 0.010 at A-UD, and its Task-A utility drops from 0.120 to 0.010 (Table 2). Similar degradation appears on LLaVA-13B (Table 2), under an alternative task order (Table 10), with 5 tasks instead of 4 (Table 12), and on a general benchmark such as TruthfulQA for the baselines (Table 14 also shows more).

Core method - the LUMoE is introduced in Section 5: To address this, the paper proposes LUMoE, which trains task-specific LoRA refusal adapters and uses a router to decide whether an input should be sent to one of these adapters or to the untouched original base MLLM (Section 5.2; Appendix B.3). The router is implemented with GLM-4V-Plus and operates by first judge the class and then choosing an entity from the candidate entity list for that class. On the reported benchmark metrics, LUMoE achieves near-saturated forget and utility scores throughout the sequence.

The authors provide source codes. Recommendation: Code demonstrates feasibility but might need some work to reach community reproduction. Authors should provide: (1) complete config files, (2) setup instructions with pinned dependencies, (3) example commands, (4) optionally relevant model weights.

**Compliance With Llm Reviewing Policy:**

Affirmed.

**Final Justification:**

I view this as a valuable paper on an important and practical problem: lifelong unlearning for MLLMs. The benchmark contribution is meaningful, and the empirical finding that standard unlearning baselines suffer severe cumulative degradation is useful and well supported by the experiments. The paper is also generally clear and well organized.

Thanks for the rebuttal and the additional analyses. In Answer 4, the human-evaluation result is helpful. It would be even stronger with a few more details, such as annotator information, evaluation instructions, and some measure of agreement/correlation, so that readers can better judge the strength of this validation.

Overall, the rebuttal helps clarify main questions. Please add the relevant analyses into the final paper.

**Key Questions For Authors:**

1. Given that MLUBench apply forgetting via suppression/refusal, and LUMoE preserves an intact base model while routing matched queries to refusal adapters (Section 5.2, B.3), could you clarify whether LUMoE should be interpreted as behavioral unlearning / non-disclosure rather than parameter-level erasure? Some additional evidence would also help broader readers see these interpretations straightforward.

2. Can you provide more direct evidence for the multimodal-alignment hypothesis?
Table 1 is suggestive, but not definitive. Do you have alignment-specific diagnostics, representation drift analyses, or projector/vision-language interaction measurements that support this claim more directly?

3. How well do the LLM-judge scores align with human judgment on a sampled subset? This is especially important for borderline cases involving partial refusals, near-miss factual answers, or generic-but-noncommittal responses.

**Limitations:**

Yes

**Strengths And Weaknesses:**

- Soundness

Strength:

The paper addresses a practical and important problem.
The move from one-shot unlearning to sequential deletion requests is realistic, and this matters especially for privacy/copyright-sensitive multimodal systems. The paper clearly formulates the lifelong setting and motivates why cumulative effects should be studied rather than only single-step unlearning.

Weakness:

The empirical degradation of the baselines is well supported by Table 2, Table 10, Table 12, and Table 14. However, the paper’s stronger claims about unlearning and multimodal alignment are not fully supported by the presented evidence.

The forget-quality metric (GPT Rejection Score) strongly rewards refusal/abstention. It defines a high-quality refusal as the best forgetting outcome. That is a defensible operational choice, but it conflates 'the model no longer reveals the knowledge' with 'the knowledge has been unlearned'. This is particularly favorable to LUMoE, whose mechanism is explicitly designed to route matched queries to refusal behavior.

- Presentation

The paper is generally well organized and easy to follow.
The narrative flows cleanly from problem setup (Sec. 3), benchmark construction (Sec. 4), method (Sec. 5), experiments (Sec. 6), and appendices. The appendices include many details that help readers understand the pipeline, including the judge prompts (Appendix C.1-C.2), benchmark question variants (A.4) etc.

The main empirical phenomenon - severe cumulative degradation for standard baselines - is convincingly demonstrated.

- Significance

Strength:

The benchmark contribution is meaningful and probably the strongest part of the paper.
The paper has comprehensive and clear technical results. In Table 2, the baseline methods collapse on both forget quality and utility across two model sizes. The same pattern is shown again under a different task order (Table 10), with more tasks (Table 12), and even on an unrelated benchmark for the baselines (Table 14). Figure 1b also provides a qualitative example of output degeneration after repeated GD updates. I find this empirical observation useful.


- Originality

The benchmark is novel and worthwhile. The method is simple, but the combination of task-specific refusal adapters with a multimodal routing mechanism is a practical design for this setting. The core method is easy to understand and has high potential been applied by other people.

---

> ### Author Rebuttal · Authors · 2026-03-30
>
> # Response to reviewer DtTC
>
> We are glad that the reviewer found the studied problem practical and important. We would like to respond to each detailed point individually.
>
> > 1. However, the paper’s stronger claims about unlearning and multimodal alignment are not fully supported by the presented evidence. Can you provide more direct evidence for the multimodal-alignment hypothesis? Table 1 is suggestive, but not definitive. Do you have alignment-specific diagnostics, representation drift analyses, or projector/vision-language interaction measurements that support this claim more directly?
>
> To provide direct evidence for the alignment claim, we measured the **Modality Gap** (L2 distance between the visual feature centroid and the language feature centroid) between vision and language representations. **A smaller modality gap means better alignment.** As shown in the table below, after unlearning, **the Modality Gap on four tasks (A to D) enlarges consistently**.
>
> ### Table: Representation drift analysis on Qwen3-VL-4B-Instruct.
> | Task    | Modality Gap (original model)   | Modality Gap (unlearned model)   | Δ Gap    |
> |--------|------|----------|----------|
> | TaskA      | 20.727      | 22.353     | +1.626  |
> | TaskB       | 19.081     | 20.067     | +0.987  |
> | TaskC      | 17.372      | 18.904     | +1.532  |
> | TaskD    | 18.522      | 19.785     | +1.263  |
>
> > 2. The forget-quality metric (GPT Rejection Score) strongly rewards refusal/abstention. It defines a high-quality refusal as the best forgetting outcome. That is a defensible operational choice, but it conflates 'the model no longer reveals the knowledge' with 'the knowledge has been unlearned'.
>
> Yes, the metrics we proposed (forget quality and model utility) are more inclined toward making the model refuse to answer. However, we believe that a response that fails to reject a question may either be a hallucination or reflect the factual knowledge of the unlearning entity, whereas a high-quality refusal effectively prevents both scenarios. Therefore, our metrics are reasonable.
>
> > 3. Could you clarify whether LUMoE should be interpreted as behavioral unlearning / non-disclosure rather than parameter-level erasure? Some additional evidence would also help broader readers see these interpretations straightforward.
>
> Yes, the LUMoE method is more like a behavioral unlearning method. However, to verify the effectiveness of the unlearning, we conducted **jailbreak attack experiments** on LUMoE in **Section 6.4 (Ablation Studies)**. The results show that LUMoE exhibits strong robustness against jailbreak attacks. Therefore, its unlearning effect is robust.
>
> > 4. How well do the LLM-judge scores align with human judgment on a sampled subset? This is especially important for borderline cases involving partial refusals, near-miss factual answers, or generic-but-noncommittal responses.
>
> We thank the reviewer for the suggestion of human judgment. Following your advice, we selected the **Forget quality results of the GA method on LLaVA-7B as a subset**. We conducted the human evaluation and compared it with the LLM evaluation results presented in the main text. The results are shown in the table below.
>
> ### Table: Analysis of the alignment between the LLM judge and the human judge.
> | Method  | A-UA | A-UB | A-UC | A-UD | B-UB | B-UC | B-UD | C-UC | C-UD | D-UD |
> |--------|----------------|-------|-------|-------|-------|-------|-------|-------|-------|-------|
> | LLM (GPT-4o)  | 0.380 | 0.195 | 0.035 | 0.010 | 0.220 | 0.130 | 0.070 | 0.185 | 0.075 | 0.060 |
> | Human  | 0.450 | 0.255 | 0.015 | 0.000 | 0.205 | 0.155 | 0.010 | 0.180 | 0.050 | 0.050 |
>
> According to the results, human evaluation is highly consistent with the LLM evaluation.

---

> > ### Author Rebuttal · Reviewer_DtTC · 2026-04-02
> >
> > Thanks for the rebuttal and the additional analyses.
> > In Answer 4, the human-evaluation result is helpful. It would be even stronger with a few more details, such as annotator information, evaluation instructions, and some measure of agreement/correlation, so that readers can better judge the strength of this validation.
> > Overall, the rebuttal helps clarify main questions. I would like to keep my overall score at 4 (weak accept), not changing sub-scores, and I am increasing my confidence from 3 to 4.

---

> > > ### Author Response · Authors · 2026-04-03
> > >
> > > Dear Reviewer DtTC,
> > >
> > > Thank you for your constructive feedback and invaluable comments, and for raising the confidence to 4.
> > >
> > > We are glad that our rebuttal helped clarify the main questions. To further strengthen the human evaluation in Answer 4, we provide the following details:
> > >
> > > **Annotators:** 2 computer-science PhD students with sufficient expertise in machine unlearning.
> > >
> > > **Evaluation Instructions:** Identical to the prompt used for the LLM judge to ensure fairness and consistency.
> > >
> > > **Agreement Protocol:** We only count a sample when both annotators agree. In case of disagreement, they discuss until reaching a consensus.
> > >
> > > We will include these details in the revised manuscript to make the human validation fully transparent and reproducible.
> > >
> > > Best regards,
> > >
> > > Authors

---

### Official Review · Reviewer_swox · 2026-03-13

**Soundness:** 3
**Presentation:** 3
**Significance:** 3
**Originality:** 3
**Overall Recommendation:** 4
**Confidence:** 4

**Summary:**

This paper proposed a benchmark for continual unlearning for MLLM. The authors curated VQA pairs based on the factual knowledge of widely known real-world entities (9 entity types) and constructed 4 tasks. Sequentially, the authors proposed an effective baseline method LUMoE to avoid alignment disruption. LUMoE trains lora adapters for each unlearning task and uses another commercial MLLM for routing the input to corresponding adapter. The experimental results are comprehensive.

**Compliance With Llm Reviewing Policy:**

Affirmed.

**Final Justification:**

In the authors' rebuttal, they provide detailed experimental results to address my concerns. Thus, I maintain my positive scores.

**Key Questions For Authors:**

1. See Weakness 2 and 3
2. Could the router be replaced by open-sourced models or smaller models?

**Limitations:**

Yes

**Strengths And Weaknesses:**

Strengths:
1. The studied problem: lifelong unlearning for MLLM, is crucial and practical. And the paper included a benchmark and baseline method.
2. The datasets are from known concepts and factual knowledge. The models to evaluate do not need to finetune. This is different from previous benchmarks and offers an alternative setting.
3. The experiments are comprehensive, including various ablations and robustness experiments. The results show the proposed LUMoE is robust to number of tasks, quality of the router model and jailbreak.

Weaknesses:
1. The proposed LUMoE method is not very clear. Every input must be fed to an additional MLLM for entity extraction and matching. This introduces additional latency.
2. The experiment setup is not very clear. It said that MLLMs unlearn all tasks in the sequence order of TaskA, TaskB, TaskC, and TaskD. But why the experiments show results of {A-UA A-UB A-UC A-UD B-UB B-UC B-UD C-UC C-UD D-UD} rather than {A-UA, B-UA, C-UA, D-UA; A-UA+B, B-UA+B, C-UA+B, D-UA+B; …} (i.e. evaluate all 4 tasks after each unlearn task)? When unlearning task A (before unlearning task B), the impact to ability on B is worth evaluating.
3. It’s suggested to describe how the Task Matching is implemented. Especially, how to find whether the entity is within the forget information set? Does the model need to cord and maintain an additional forget information set for each unlearning task?

---

> ### Author Rebuttal · Authors · 2026-03-30
>
> # Response to reviewer swox
> We are glad the reviewer found the studied problem crucial and practical. We would like to respond to each detailed point individually.
>
> > 1. The proposed LUMoE method is not very clear. Every input must be fed to an additional MLLM for entity extraction and matching. This introduces additional latency.
>
> We understand the reviewer's concerns over the latency and practicality. However, the computation costs are **highly manageable**. The additional MLLM is employed via API calling, which effectively offloads the local computing cost. In addition, given the difficulty of multimodal understanding, the router must decide when and how to activate an unlearning adapter based on both the visual and textual input. Therefore, a powerful additional MLLM router is not a flaw but a requirement and strength.
>
> > 2. The experiment setup is not very clear. It said that MLLMs unlearn all tasks in the sequence order of TaskA, TaskB, TaskC, and TaskD. But why the experiments show results of {A-UA A-UB A-UC A-UD B-UB B-UC B-UD C-UC C-UD D-UD} rather than {A-UA, B-UA, C-UA, D-UA; A-UA+B, B-UA+B, C-UA+B, D-UA+B; …} (i.e. evaluate all 4 tasks after each unlearn task)? When unlearning task A (before unlearning task B), the impact to ability on B is worth evaluating.
>
> We adopt the experimental setup of A-UA, A-UB, A-UC, A-UD, ... to **align with our definition of MLLM Lifelong Unlearning in Section 3.2**: "The objective of MLLM lifelong unlearning is to minimize the MLLM’s performance degradation on previously unlearned tasks." This definition mainly focuses on mitigating cumulative degradation (i.e., stability) on tasks that have already been unlearned. Therefore, in the main text, we evaluate the model's performance on a task only after it has been unlearned.
>
> However, we **fully agree with the reviewer's point** that: When unlearning task A (before unlearning task B), the impact on the model's ability regarding task B is worth evaluating.
> Following your advice, we **used the GA and GD methods to unlearn task A on Qwen3-VL-4B-Instruct**. Then, we **evaluated the model on the retain sets of all four tasks** and compared the results with the model's performance before unlearning task A. This was done to investigate the influence of the unlearning method on data that the model has not yet been instructed to unlearn. The results are in the following table.
>
> ### Table: Qwen3-VL-4B-Instruct's performance on all four tasks' retain set.
> | Method | State | Task A | Task B | Task C | Task D |
> |--------|----------------|-------|-------|-------|-------|
> |  | Before unlearn task A | 0.95 | 0.97 | 0.94 | 0.98 |
> | GA | After unlearn task A but before other tasks | 0.27 | 0.60 | 0.68 | 0.58 |
> | GD | After unlearn task A but before other tasks | 0.32 | 0.72 | 0.75 | 0.83 |
>
> According to the results, after unlearning task A, the model exhibited a significant performance drop on the retain sets of all four tasks. This demonstrates that the unlearning method undermines the models' performance on unrelated tasks.
>
> > 3. It’s suggested to describe how the Task Matching is implemented. Especially, how to find whether the entity is within the forget information set? Does the model need to cord and maintain an additional forget information set for each unlearning task?
>
> We thank the reviewer for this suggestion. To implement Task Matching, we maintain a lightweight list of entity names for each unlearned task. After each unlearning task, we store the names of the forgotten entities in that task's list. Once the router extracts the entity name from the input, it compares this name against existing task lists. If a match is found, the corresponding LoRA adapter is activated; otherwise, the original model is used. Note that each list only stores entity names (strings), so the storage and computational overhead is negligible.
>
> > 4. Could the router be replaced by open-sourced models or smaller models?
>
> Yes, the router can be replaced with other smaller open-source models. To verify the effectiveness of smaller open-source models, we used **Qwen3-VL-4B-Instruct and Qwen3-VL-8B-Instruct** as the router. The results are shown in the table below.
>
> ### Table: Open-sourced models router performance.
> | Model | Metrics | Task A | Task B | Task C | Task D |
> |--------|----------------|-------|-------|-------|-------|
> | Qwen3-VL-4B-Instruct | Forget Quality | 1.000 | 0.910 | 0.980 | 0.960 |
> |  | Model Utility | 0.930 | 0.640 | 0.940 | 0.910 |
> | Qwen3-VL-8B-Instruct | Forget Quality | 1.000 | 0.880 | 0.990 | 0.960 |
> |  | Model Utility | 0.930 | 0.730 | 0.940 | 0.910 |
>
> According to the results, it can be seen that strong open-source models can also achieve good performance.

---

### Official Review · Reviewer_3sqs · 2026-03-14

**Soundness:** 2
**Presentation:** 3
**Significance:** 3
**Originality:** 3
**Overall Recommendation:** 4
**Confidence:** 4

**Summary:**

The paper studies lifelong unlearning for multimodal large language models (MLLMs), where deletion requests arrive sequentially over time and the model must repeatedly forget targeted multimodal knowledge while preserving retained knowledge and general capability. To support this setting, the authors introduce MLUBench, a benchmark with 127 real-world entities across 9 classes, 5,105 images, and 15,414 VQA pairs, organized into sequential unlearning tasks with prompt variants for robustness evaluation. Using this benchmark, the paper argues that standard unlearning baselines suffer severe cumulative degradation and that MLLM lifelong unlearning has a distinctive challenge relative to unimodal models: repeated updates can disrupt multimodal alignment. To address this, the paper proposes LUMoE, a modular method that trains task-specific LoRA refusal adapters and uses a gate/router to dispatch inputs either to the appropriate adapter or to the untouched base model. Experiments on LLaVA-7B and LLaVA-13B show strong improvements over standard sequential unlearning baselines on the proposed benchmark, along with supporting ablations on router choice, prompt variants, task order, number of tasks, jailbreak robustness, and downstream general benchmarks.

**Compliance With Llm Reviewing Policy:**

Affirmed.

**Final Justification:**

I appreciate the authors’ candid acknowledgment that the current comparison is not fully apples-to-apples, and I agree that making this distinction more explicit in the paper would improve the presentation and framing. However, their responses do not fully resolve my main remaining concern, because it does not add new empirical evidence or a closer modular baseline comparison. As a result, while the rebuttals make the paper’s positioning more honest, it does not change my overall assessment enough to raise my recommendation from Weak Accept to Accept.

**Key Questions For Authors:**

1) How much of LUMoE’s gain comes from the external router, as opposed to the unlearning adapters themselves?
Since the method relies on GLM-4V-Plus for entity extraction and routes retain queries directly to the untouched base model, I would like to see comparisons to much simpler routing baselines (e.g., exact string matching, a lightweight classifier, or heuristic entity lookup) and explicit router accuracy numbers. If a simple router achieves similar results, that would materially change my view of the novelty and significance of the method.

2) Can you compare LUMoE against stronger modular baselines, such as other adapter-isolation or continual-learning expert methods, rather than only repeated full-model unlearning baselines?
This would improve the fairness of the comparison and could change my soundness evaluation.


3) Can you provide stronger evidence for the multimodal-alignment claim beyond the freeze-one-side experiments?
The Unlearn-LLM-Only and Unlearn-Vision-Only experiments are suggestive, but they do not fully isolate multimodal alignment as the fundamental cause of degradation. Additional evidence—e.g., alignment-sensitive diagnostics, stronger PEFT controls, or comparisons to other modular baselines—would strengthen the core scientific claim. A strong response here would improve both my soundness and significance assessments.

4) Can you report lifelong-unlearning experiments on a non-LLaVA family such as Qwen, not just pre-unlearning accuracy?
The benchmark is presented as broadly applicable, but the main lifelong-unlearning experiments are only on LLaVA-7B and 13B, while Qwen is only used for pre-unlearning accuracy checks. Results on a second model family would substantially improve confidence in the benchmark’s generality.

5) Can you include a cost-normalized comparison to other modular or continual-learning baselines?
LUMoE stores multiple adapters and uses an external commercial router, which changes both the computational and systems assumptions relative to the baselines. A more apples-to-apples comparison—either to adapter-based sequential baselines or to a cost-normalized version of the current baselines—would make the empirical comparison more convincing.

**Limitations:**

No. The impact statement is very brief and almost entirely positive. It does not meaningfully discuss technical limitations, benchmark biases, or misuse risks.

To improve, the authors should explicitly discuss at least the dependence on a closed-source external router and GPT-based judges, the fact that LUMoE may function more like routing plus refusal than standard model unlearning, the benchmark bias introduced by filtering examples to those already answered correctly by LLaVA models, the scalability limits of storing many adapters over long horizons, and the narrower scope of the benchmark as an entity-centric forgetting test rather than a full test of erased multimodal knowledge. They should also address negative-use concerns more directly, including censorship or selective suppression of information, and clarify the copyright/data-provenance implications of using publicly scraped images.

**Strengths And Weaknesses:**

* Soundness

(+) The paper tackles a timely and practically relevant problem. Most multimodal unlearning work
has focused on one-shot unlearning or narrower settings, whereas this paper studies sequential /
lifelong removal requests, which is closer to realistic deployment. That is a meaningful shift in problem framing, and the benchmark itself is a potentially valuable contribution. The scale and diversity of MLUBench are also solid relative to prior MLLM unlearning benchmarks discussed
in the paper, such as MMUBench, FIUBench, and MLLMU-Bench. The benchmark covers more entity types and is explicitly organized for sequential evaluation, which is a real strength.

(+) The empirical finding that standard unlearning baselines degrade severely under repeated tasks is interesting and, at least directionally, convincing. Table 2 shows catastrophic drops for GA, KL, GD, and NPO across both LLaVA-7B and 13B, while LUMoE remains high on both forget
quality and utility. Even if one questions whether the numbers are too clean, the broader point
that repeated parameter updates are unstable in this setting is plausible and important.

(-) The main soundness issue is that LUMoE is not really comparable to the baselines as a single-model unlearning method. Its key mechanism is to use an external commercial router (GLM-4V-Plus) to extract entities and route forget requests to task-specific refusal adapters while sending retained inputs directly to the original untouched base MLLM. This means its near-perfect utility is, to a large extent, achieved by construction: retained questions bypass the unlearned system entirely. That makes LUMoE closer to a routing/access-control system than to a standard unlearning update on one model, so the headline comparison to sequential weight-editing baselines is not apples-to-apples.

(-) The multimodal-alignment claim is not fully established at the level the text suggests. The
paper says it “proves” the uniqueness of MLLM lifelong unlearning via Unlearn-LLM-Only and Unlearn-Vision-Only experiments, but those results only show that updating either side alone
damages performance. That is consistent with alignment fragility, but it does not isolate
alignment as the sole cause; repeated catastrophic interference within either subsystem could also explain some degradation. The evidence is suggestive, not definitive. The language should be softened from “prove” to “support” or “indicate.”

(-) There are also some benchmark/evaluation caveats. The dataset is filtered to retain only examples that both LLaVA-7B and LLaVA-13B already answer correctly, and the main lifelong-unlearning experiments are also on those same two LLaVA models. That is reasonable for ensuring pre-unlearning knowledge, but it also means the benchmark is partially tailored to the evaluated family. In addition, forget quality and utility are both judged by GPT-4o-based scoring rather than a retraining-based gold standard, so the evaluation measures high-quality refusal/correctness on benchmark queries rather than true deletion in a stronger machine-unlearning sense.


*Presentation
(+) The paper is structurally clear. The benchmark motivation, problem formulation, method, and experiments are easy to follow, and the core message comes through cleanly: existing sequential unlearning baselines degrade badly, and the authors advocate isolating task-specific changes rather than repeatedly editing one model. Figures 1 and 2 are especially helpful for conveying the setup.

(-) Reproducibility is decent but not ideal. Some practical details are present, but the dependence on commercial components (GLM-4V-Plus router, GPT-4o judge) weakens full reproducibility, and the scalability discussion in the appendix is fairly brief relative to how central the routing design is to the method.


* Significance
(+) The problem is important. In real deployments, deletion requests do arrive over time, and multimodal systems add extra complications beyond text-only models. A benchmark specifically for lifelong multimodal unlearning is therefore a useful contribution, and the paper’s empirical evidence that standard baselines degrade badly could be valuable for future work in privacy, copyright, and data governance for MLLMs.

(+) I also think the benchmark itself could be useful even if one is not fully convinced by LUMoE. The task-sequence design, variant prompts, and downstream checks provide a reasonable starting point for studying sustainability of unlearning rather than just one-shot forgetting.

(-) Since LUMoE largely avoids catastrophic forgetting by not repeatedly editing the original model, its success may reflect a practical workaround rather than a substantive advance in machine unlearning itself. That is still useful, but it is more of a modular engineering design than a decisive step toward principled lifelong unlearning.


* Originality
(+) The benchmark contribution is reasonably original. I am not aware of many prior MLLM unlearning benchmarks that are this explicitly focused on sequential/lifelong requests at this scale, and the paper’s framing around cumulative degradation is useful. The idea of highlighting multimodal alignment as a key pressure point is also a meaningful perspective.

(-) Still, the method is not a major conceptual leap. It is essentially a modular routing architecture built from familiar components, and much of its effectiveness may stem from avoiding the hardest part of the problem rather than solving it directly. So I would rate originality as moderate: meaningful and nontrivial, but not especially high.

---

> ### Author Rebuttal · Authors · 2026-03-30
>
> # Response to reviewer 3sqs
>
> > 1. Can you compare LUMoE against stronger modular baselines, such as other adapter-isolation methods?
>
> Currently, MLLM lifelong unlearning is still a new research area. To the best of our knowledge, there are no similar adapter-isolation continual MLLM unlearning baselines available. Therefore, it is hard for us to perform direct comparisons with other adapter-isolation methods.
>
> > 2. Can you provide stronger evidence for the multimodal-alignment claim beyond the freeze-one-side experiments?
>
> To provide direct evidence for the alignment claim, we measured the **Modality Gap** (L2 distance between the visual feature centroid and the language feature centroid) between vision and language representations. **A smaller modality gap means better alignment.** As shown in the table below, after unlearning, **the Modality Gap on four tasks (A to D) enlarges consistently**.
>
> ### Table: Representation drift analysis on Qwen3-VL-4B-Instruct.
> | Task    | Modality Gap (original model)   | Modality Gap (unlearned model)   | Δ Gap    |
> |----|---|----|----|
> | A    | 20.727      | 22.353     | +1.626  |
> | B    | 19.081     | 20.067     | +0.987  |
> | C    | 17.372      | 18.904     | +1.532  |
> | D    | 18.522      | 19.785     | +1.263  |
>
> > 3. There are also some benchmark/evaluation caveats. In addition, the evaluation measures high-quality refusal/correctness on benchmark queries rather than true deletion in a stronger machine-unlearning sense.
>
> We thank the reviewer for this insightful comment. We fully agree that "true deletion" is the ultimate goal of machine unlearning. However, **rigorously defining and measuring internal deletion in MLLMs remains an open challenge.** In this work, we adopt high-quality refusal as the metric because it serves as a reliable behavioral proxy: failure to refuse either leaks factual knowledge or produces hallucinations, both indicating unsuccessful unlearning. High-quality refusal effectively suppresses both scenarios.
>
> > 4. Reproducibility is decent but not ideal, and the scalability discussion in the appendix is fairly brief relative to how central the routing design is to the method.
>
> **Reproducibility:** We use the commercial MLLM (e.g., GLM-4V-Plus) for its strong performance. However, our routing and evaluation frameworks are model-agnostic. The core contribution lies in the routing design and prompt engineering. We have fully documented them and provided all exact prompts. Our tests in **response 6** also show that strong open-source models work well as alternatives.
>
> **Scalability:** We appreciate the reviewer's suggestion. Scaling routing to millions of requests is an industry-wide challenge. Our current design focuses on modular efficiency. In the revised manuscript, we have added a more detailed discussion, including potential strategies such as batch processing.
>
> > 5. Since LUMoE largely avoids catastrophic forgetting by not repeatedly editing the original model, its success may reflect a practical workaround rather than a substantive advance in machine unlearning itself.
>
> Although LUMoE is a relatively simple method, it achieves strong performance and validates the effectiveness of the isolation idea. In addition, this study mainly focuses on the benchmark and performance analysis of existing methods on MLLMs, rather than developing a new unlearning algorithm.
>
> > 6. How much of LUMoE’s gain comes from the external router, as opposed to the unlearning adapters themselves? I would like to see comparisons to much simpler routing baselines (e.g., exact string matching) and explicit router accuracy numbers.
>
> Both the router and the unlearning adapters are important, as supported by the following empirical study. Due to the complexity of multimodal reasoning, we cannot directly adopt simple methods such as string matching. However, to examine the effectiveness of simpler routers, we used the smaller **Qwen3-VL-4B-Instruct and Qwen3-VL-8B-Instruct as the routers**. The results and router accuracy are shown below.
>
> ### Table: Open-sourced model router performance and accuracy.
> | Model | Metrics | Task A | Task B | Task C | Task D |
> |---|---|--|---|---|--|
> | Qwen3-VL-4B-Instruct | Forget Quality | 1.000 | 0.910 | 0.980 | 0.960 |
> | Accuracy: 97.1% | Model Utility | 0.930 | 0.640 | 0.940 | 0.910 |
> | Qwen3-VL-8B-Instruct | Forget Quality | 1.000 | 0.880 | 0.990 | 0.960 |
> | Accuracy: 98.0% | Model Utility | 0.930 | 0.730 | 0.940 | 0.910 |
>
> According to the results, strong open-source routers can also achieve good performance.
>
> > 7. Can you report lifelong-unlearning experiments on a non-LLaVA family such as Qwen, not just pre-unlearning accuracy?
>
> Following your advice, we **conducted experiments on the Qwen3-VL-4B-Instruct model**. The detailed results can be found in **response 2 for reviewer PqdF**. According to the results, baselines exhibit cumulative degradation, while LUMoE still demonstrates strong performance.

---

> > ### Author Rebuttal · Reviewer_3sqs · 2026-04-02
> >
> > The rebuttal improves my confidence in the paper’s practical value, especially as a benchmark-and-analysis contribution, but it does not fully resolve my main concerns. In particular, several responses provide additional supporting evidence or clarifications, yet the central comparability issue remains: LUMoE is still better understood as a routing-and-isolation workaround than as an apples-to-apples lifelong unlearning method, so the comparison to standard sequential unlearning baselines is still not fully fair. The added experiments on router choice, modality gap, and Qwen generalization are helpful, but they only partially address the concerns about mechanism, generality, and attribution of gains.
> >
> > For example, the first rebuttal point did not really resolve. Saying there are no directly comparable adapter-isolation lifelong MLLM unlearning baselines is understandable, but it does not answer the core fairness concern. The issue was not only “is there an exact prior method,” but whether LUMoE should be compared against other modular / continual / routing-style alternatives rather than only repeated full-model unlearning baselines. The rebuttal does not provide such comparisons.
> >
> > Overall, the rebuttal strengthens the paper, but not enough for me to raise my recommendation beyond Weak Accept.

---

> > > ### Author Response · Authors · 2026-04-03
> > >
> > > Dear Reviewer 3sqs,
> > >
> > > We really appreciate your constructive feedback and for clearly articulating the remaining concern.
> > >
> > > Since MLLM lifelong unlearning is still a new area, **to the best of our knowledge, no prior work has proposed a directly comparable adapter-isolation / routing-style method for this exact setting.** We thank you for your understanding that we cannot make a direct comparison with other adapter-isolation lifelong MLLM unlearning baselines. We agree that the current comparison may not be fully fair. LUMoE is a modular routing-and-isolation approach, while the baselines are repeated single-model weight updates. However, this difference is indeed central to our work, as one of our core findings is that repeated full-model updates severely damage MLLMs' performance.
> > >
> > > Nevertheless, we **acknowledge that a fair comparison to other modular-style alternatives is needed and would strengthen our paper**.
> > >
> > > **In the revised manuscript**, we will:
> > > 1. Expand the related work and discussion sections to explicitly compare LUMoE with the closest routing style baselines from relevant fields (e.g., routing style methods for LLM unlearning).
> > >
> > > 2. Add a dedicated subsection highlighting the core advantages of our LUMoE as well as its limitations relative to traditional single-model update unlearning methods.
> > >
> > > We hope the above information will address your remaining concern about the comparison.
> > >
> > > Best regards,
> > >
> > > Authors

---

### Decision · Program_Chairs · 2026-04-30

**Decision:**

Accept (regular)

**Comment:**

The paper generally receives positive feedback from the reviewers. Reviewers (DtTC, 3sqs, swox) acknowledged that the paper addresses a practical and important problem. In fact, MLUBench presents a novel and useful benchmark for lifelong unlearning in MLLMs. On the other hand, reviewers have concerns about the newly proposed method LUMoE, for example regarding its clarity and how it compares with prior methods.

I read the paper myself and am leaning toward acceptance (borderline acceptance) because I view the key contribution of the paper as the benchmarking of lifelong unlearning in MLLMs, while LUMoE is mainly included to illustrate the limitations of existing unlearning methods and provide a practical reference point for the benchmark. To make the paper stronger, I think the authors should make this positioning more explicit in the final version. In particular, the paper would benefit from framing MLUBench as the primary contribution and presenting LUMoE more clearly as a practical baseline or modular solution rather than as a fully comparable end-to-end unlearning method. The discussion should also more directly acknowledge the caveats raised by the reviewers, including the dependence on routing, the distinction between refusal-based behavior and true unlearning, and the additional latency or complexity introduced by the method. With these clarifications, I believe the paper would make a useful contribution to the community.